# Rethinking the role of transport and photochemistry in regional ozone pollution: Insights from ozone concentration and mass budgets

Kun Qu[1,2,3], Xuesong Wang[1,2], Xuhui Cai[1,2], Yu Yan[1,2], Xipeng Jin[1,2], Mihalis Vrekoussis[3,4,5], Maria Kanakidou[3,6], Guy Brasseur[7,8], Jin Shen[9], Teng Xiao[1,2], Limin Zeng[1,2], and Yuanhang Zhang[1,2,10,11]

[1]State Key Joint Laboratory of Environmental Simulation and Pollution Control, College of Environmental Sciences and Engineering, Peking University, Beijing 100871, China
[2]International Joint Laboratory for Regional Pollution Control, Ministry of Education, Beijing, 100816, China
[3]Laboratory for Modeling and Observation of the Earth System (LAMOS), Institute of Environmental Physics (IUP), University of Bremen, Bremen, Germany
[4]Center of Marine Environmental Sciences (MARUM), University of Bremen, Germany
[5]Climate and Atmosphere Research Center (CARE-C), The Cyprus Institute, Cyprus
[6]Environmental Chemical Processes Laboratory, Department of Chemistry, University of Crete, Heraklion, Greece
[7]Max Planck Institute for Meteorology, Hamburg, Germany
[8]National Center for Atmospheric Research, Boulder, Colorado, USA
[9]State Key Laboratory of Regional Air Quality Monitoring, Guangdong Key Laboratory of Secondary Air Pollution Research, Guangdong Environmental Monitoring Center, Guangzhou 510308, China
[10]Beijing Innovation Center for Engineering Science and Advanced Technology, Peking University, Beijing 100871, China
[11]CAS Center for Excellence in Regional Atmospheric Environment, Chinese Academy of Sciences, Xiamen 361021, China

*Correspondence to*: Xuesong Wang (xswang@pku.edu.cn) and Yuanhang Zhang (yhzhang@pku.edu.cn)

**Abstract.** Understanding the role of transport and photochemistry is essential to mitigate tropospheric ozone ($O_3$) pollution within a region. In previous studies, the $O_3$ concentration budget has been widely used to determine the contributions of two processes to the variations of $O_3$ concentrations. These studies often conclude that local photochemistry is the main cause of regional $O_3$ pollution; however, they fail to explain why $O_3$ in a targeted region is often primarily derived from $O_3$ and/or its precursors transported from the outside regions as reported by many studies of $O_3$ source apportionment. Here, we present a method to calculate the hourly contributions of $O_3$-related processes to the variations of not only the mean $O_3$ concentration, but also the total $O_3$ mass (the corresponding budgets are noted as the $O_3$ concentration and mass budget, respectively) within the atmospheric boundary layer (ABL) of the concerned region. Based on the modelling results of WRF-CMAQ, the two $O_3$ budgets were applied to comprehensively understand the effects of transport and photochemistry on the $O_3$ pollution over the Pearl River Delta (PRD) region in China. Quantified results demonstrate the different role of transport and photochemistry when comparing the two $O_3$ budgets: Photochemistry drives the rapid increase of $O_3$ concentrations during the day, whereas transport, especially vertical exchange through the ABL top, controls both rapid $O_3$ mass increase in the morning and decrease in the afternoon. The diurnal changes of the transport contributions in the two $O_3$ budgets highlight the influences of the ABL diurnal cycle and regional wind fields on regional $O_3$ pollution. Through high contributions to the $O_3$ mass increase in the morning, transport determines that most $O_3$ in the PRD originates from the global background and emissions outside the region. However, due to the simultaneous rapid increase of ABL volumes, this process only has a relatively limited effect on $O_3$ concentration increase compared to photochemistry, and transport effect on the regional sources of $O_3$ cannot be illustrated by

the $O_3$ concentration budget. For future studies targeting $O_3$ and other secondary pollutants with moderately long atmospheric
lifetimes (e.g., fine particulate matter and some of its components), insights from both concentration and mass budgets are
required to fully understand the role of transport, chemistry and other related processes.

## 40   1 Introduction

Since first recognized as a key contributor to the Los Angeles smog, tropospheric ozone ($O_3$) pollution has received
considerable attention in many highly populated areas in the world (Fishman et al., 2003; Schultz et al., 2017; Fleming et al.,
2018; Fowler et al., 2020). Exposure to $O_3$ threatens crop yields, ecosystems and human health, resulting in increased
mortality and economic losses (Mills et al., 2013; Ainsworth, 2017; Zhang et al., 2019). In addition, $O_3$ contributes to global
warming not only directly as a greenhouse gas, but also indirectly by damaging plants and suppressing land carbon sinks
(Sitch et al., 2007; Naik et al., 2021). To address these detrimental effects, efforts have been undertaken to reduce $O_3$ levels
in polluted regions. However, since $O_3$ is a secondary pollutant produced in the atmosphere by complex non-linear
chemistry, the abatement of $O_3$ pollution is a challenging task.

As a prerequisite to effectively control $O_3$ pollution, firstly, it is imperative to understand the effects of $O_3$-related processes
on the abundance of $O_3$ in the atmosphere. High $O_3$ concentrations within a region are often attributed to daytime
photochemical production from $O_3$ precursors, i.e. $NO_x$ (= $NO + NO_2$) and volatile organic compounds (VOCs), under
sunlight. Due to the short lifetime of $O_3$ precursors (several hours for $NO_x$ and reactive VOCs (Liu et al., 2016; Seinfeld and
Pandis, 2016; Laughner and Cohen, 2019)), it is generally believed that $O_3$ photochemistry is mainly linked to the
contributions of local emissions in polluted regions. On the other hand, since $O_3$ itself has a moderately long atmospheric
lifetime of 20-30 days (Stevenson et al., 2006; Bates and Jacob, 2019), transport processes in the atmosphere, including
horizontal transport (mainly advection) and vertical exchange through the top of the atmospheric boundary layer (ABL), may
also considerably contribute to regional $O_3$ pollution (Myriokefalitakis et al., 2016). Specifically, through vertical exchange,
$O_3$ in the residual layer and/or free atmosphere is entrained into the ABL and involved in the ABL mixing after sunrise,
leading to rapidly increasing $O_3$ concentrations near the surface (Kaser et al., 2017; Hu et al., 2018; Zhao et al., 2019).
Although $O_3$ produced from local emissions may be transported out and later recirculated back to the region, it is more likely
that transported $O_3$ is mainly derived from the emissions of $O_3$ precursors in the upwind regions, continents and even $O_3$ in
the stratosphere under the combined effect of meso-, synoptic-, large- and global-scale atmospheric movements (Massagué
et al., 2019). If photochemistry has a comparatively large influence on $O_3$, reducing local emissions is an appropriate strategy
to alleviate regional $O_3$ pollution; otherwise, it is necessary to focus on emission control in the upwind regions, aiming to
reduce transport contributions to $O_3$.

In many studies, the O₃ concentration budget was often utilized to quantify the contributions of various transport and
chemical processes to the variations of O₃ concentrations. The changes in the mean O₃ concentration within the ABL ($\langle c_{O_3}\rangle$)
can be expressed as the net contributions of all O₃-related processes (Lenschow et al., 1981; Janssen and Pozzer, 2015; Vilà-
Guerau de Arellano et al., 2015):

$$\frac{\partial \langle c_{O_3}\rangle}{\partial t} = -\bar{u}\frac{\partial \langle c_{O_3}\rangle}{\partial x} - \bar{v}\frac{\partial \langle c_{O_3}\rangle}{\partial y} - \frac{\partial \overline{c_{O_3}'w'}}{\partial z} + S(O_3) \tag{1}$$

where $u$, $v$ and $w$ refer to wind speeds in the x-, y- and z-direction, respectively. The right side of Eq. (1) describes the
contributions of 1) horizontal transport (advection, the first two terms), 2) vertical exchange through the ABL top
(entrainment and detrainment, the third term), 3) gas-phase chemistry, dry deposition and other processes (the term $S(O_3)$
indicates their net contributions). The O₃ concentration budget is then derived by integrating these terms over time. It enables
the identification of the processes that produce positive or negative tendencies of the O₃ concentration, and of the processes
that are most influential for regional O₃ pollution. Reported O₃ concentration budgets derived from ground-based
measurements (Su et al., 2018; Tan et al., 2018; Tan et al., 2019; Yu et al., 2020), aircraft-based mobile observations
(Lenschow et al., 1981; Trousdell et al., 2016; Trousdell et al., 2019) and Process Analysis (PA) or similar modules in
chemical transport models (Hou et al., 2014; Li et al., 2021; Yan et al., 2021) in various regions of the globe often suggest
that O₃ production through local photochemistry drives the noon-time increase of O₃ concentration, whereas transport
reduces O₃ concentration over the same period. Conclusively, photochemistry, rather than transport, plays a main role in O₃
pollution.

However, O₃ source apportionment is likely to provide different conclusions about the relative importance of transport and
photochemistry in affecting O₃ pollution. O₃ source apportionment is performed to identify the regional and/or sectoral
origins of O₃, of which the results are also used to support air pollution control (Clappier et al., 2017; Thunis et al., 2019).
Here, we only discuss the regional origins of O₃, because the contributions of sources outside the region (or emissions within
the region, defined as local emissions hereafter) provide information on the influence of transport (or photochemistry) on O₃
pollution. Previous publications often conclude that most O₃ was not derived from local emissions of O₃ precursors, but from
the global background and emissions outside the targeted regions (Guo et al., 2018; Pay et al., 2019; Liu et al., 2020). The
mixing ratios of background O₃ in various regions of the world are mostly within the range of 30-50 ppb (Reid et al., 2008
and references therein), which are sufficiently high to ensure that O₃ originates mainly from non-local sources in less
polluted regions. Since controlling background O₃ is challenging, efforts to control O₃ pollution in polluted regions with high
non-local contributions to O₃ should focus on reducing emissions in upwind regions rather than only local areas (Lelieveld et
al., 2009; Boian and Andrade, 2012; Massagué et al., 2019). One successful example is the establishment of the "Ozone
Transport Region" in the north-eastern United State by the US Environmental Protection Agency, which promotes
collaborative emission reductions among states to address inter-state O₃ transport (Novel, 1992). The above discussion
highlights the importance of transport for regional O₃ pollution, since it often plays a more prominent role than local
photochemistry. Apparently, this last statement conflicts with the conclusions derived from the $O_3$ concentration budget.
Thus, while the $O_3$ concentration budget is useful for understanding $O_3$ pollution, it may not completely illustrate the effects
of transport and photochemistry on regional $O_3$ pollution.

In the ABL of the concerned region, the mean $O_3$ concentration and total $O_3$ mass are both conserved, which means that their
variations are equal to the net contributions by various $O_3$-related processes including transport and photochemistry. These
relationships can be represented by the $O_3$ concentration budget and mass budget, respectively. Unlike the aforementioned
$O_3$ concentration budget in Eq. (1), the hourly $O_3$ mass budget, written as

$$\frac{\partial m_{O_3}}{\partial t} = -\left(\bar{u}s_x\langle c_{O_3}\rangle + \bar{v}s_y\langle c_{O_3}\rangle\right) - \overline{c_{O_3}'w'}s_z + S(O_3)V \tag{2}$$

is seldom reported ($m_{O_3}$ is the total $O_3$ mass within the ABL of the region; $s_x$, $s_y$, $s_z$ are the areas of the interfaces in the x-,
y- and z-direction, respectively; $V$ is the volume of the ABL column). Due to the varied effects of transport on $O_3$
concentration and mass, the $O_3$ mass budget differs from the $O_3$ concentration budget but is more suitable to explore the
influence of transport and photochemistry on the results of $O_3$ source apportionment (more detailed explanations are given in
Sect. 2.4). In order to comprehensively understand the role of transport and photochemistry in regional $O_3$ pollution, in the
present study, we developed a method to calculate both the $O_3$ concentration and mass budget based on the simulation results
from the Weather Research and Forecasting (WRF) and Community Multiscale Air Quality (CMAQ) models, and also
analysed, compared the results of the two regional-level $O_3$ budgets. The Pearl River Delta (PRD) region, a city cluster
located on the southeast coast of China and exposed to severe $O_3$ pollution in summer and autumn (Gao et al., 2018), was
selected as the targeted region. The tasks for this study can be summarized as follows:

*1) Development of the method to quantify the two $O_3$ budgets*
WRF-CMAQ employs the Process Analysis (PA) module to assess the contributions of $O_3$-related processes to the variations
of $O_3$ concentrations within each grid cell. However, to obtain the regional-level $O_3$ concentration and mass budgets, the
results of PA module are not sufficient. One reason is that the contribution of vertical exchange through the ABL top is not
specifically quantified in commonly used ABL parameterizations, thus requires additional calculations (Kaser et al., 2017).
Additionally, calculations based on the PA results are needed to identify the contributions of other $O_3$-related processes to
ABL-mean $O_3$ concentration as well as the results of the $O_3$ mass budget. To address this, we developed a method to quantify
the two $O_3$ budgets, of which the details are given in Sect. 2.1-2.3.

*2) Analysis and comparison of the results from the two $O_3$ budgets*
Based on the simulations of $O_3$ pollution in the PRD with the model setup introduced in Sect. 2.5, the two $O_3$ budgets were
calculated for further analyses and comparisons to reveal the role of transport and photochemistry in regional $O_3$ pollution
from a more comprehensive perspective. Relative discussions are presented in Sect. 3.

*3) Assessment of the role of transport and photochemistry in determining the regional origins of O₃*
The Brute Force Method (BFM; Clappier et al., 2017), a widely used source apportionment method, was combined with the
$O_3$ mass budget calculation to determine the contributions of emissions within and outside the PRD as well as background
sources to the $O_3$ transported into or produced by photochemistry in the region (methodology described in Sect. 2.6). The
results, as discussed in Sect. 4, reveal the impacts of transport and photochemistry in determining the regional origins of $O_3$
in the PRD, and explain why the different views on the role of two processes in regional $O_3$ pollution are suggested by the $O_3$
concentration budget and $O_3$ source apportionment studies.
**2 Methodology: O₃ budget calculations and model setup**
**2.1 The PRD grids and O₃-related processes in O₃ budgets**
The two $O_3$ budgets were calculated for the PRD, of which the grids are shown in the lower-left panel of Fig. 1. These grids
are set based on the finer modelling domain of WRF-CMAQ (details given in Sect. 2.5) and determined according to the
administrative areas of the PRD. The PRD grids with one or several interfaces with the outer regions are defined as the
border grids, and they can be further classified as the grids in the north, south, west and east borders based on their locations.
Correspondingly, the PRD grids with no interface with the outer regions are defined as the non-border grids.

Figure 1 also displays all $O_3$-related processes considered in the calculation of $O_3$ budgets here. The transport processes
include horizontal transport through the four types of borders and vertical exchange through the ABL top. For vertical
exchange, its contribution in the $O_3$ concentration budget (the third term on the right side of Eq. (1)) is quantified by (Sinclair
et al., 2010; Jin et al., 2021):

$$-\frac{\partial \overline{c_{O_3}'w'}}{\partial z} = \frac{\Delta c_{O_3}}{H}\frac{\partial H}{\partial t} + \frac{\Delta c_{O_3}}{H}\left(u_h\frac{\partial H}{\partial x} + v_h\frac{\partial H}{\partial y} - w_h\right) \tag{3}$$

where $H$ is the ABL height; $\Delta c_{O_3}$ is the difference between $O_3$ concentrations above and within the ABL; $u_h$, $v_h$ and $w_h$ are
the ABL-top wind speeds in the x, y and z-direction, respectively. The terms on the right side of Eq. (3) suggest that vertical
exchange through the ABL top, or entrainment and detrainment, is attributed to 1) the temporal changes of ABL heights ($H$)
and 2) advection perpendicular to the ABL top and its slope. For the convenience of discussion, hereafter, vertical exchanges
due to the above two dynamic processes are marked as ABLex-H and ABLex-A, respectively. The contributions of all
transport processes in the $O_3$ budgets were quantified based on meteorological parameters simulated by WRF and $O_3$
concentrations simulated by CMAQ. The basic calculations of the contributions from the above-mentioned transport
processes in the $O_3$ mass and concentration budgets are separately introduced in the following two sections.

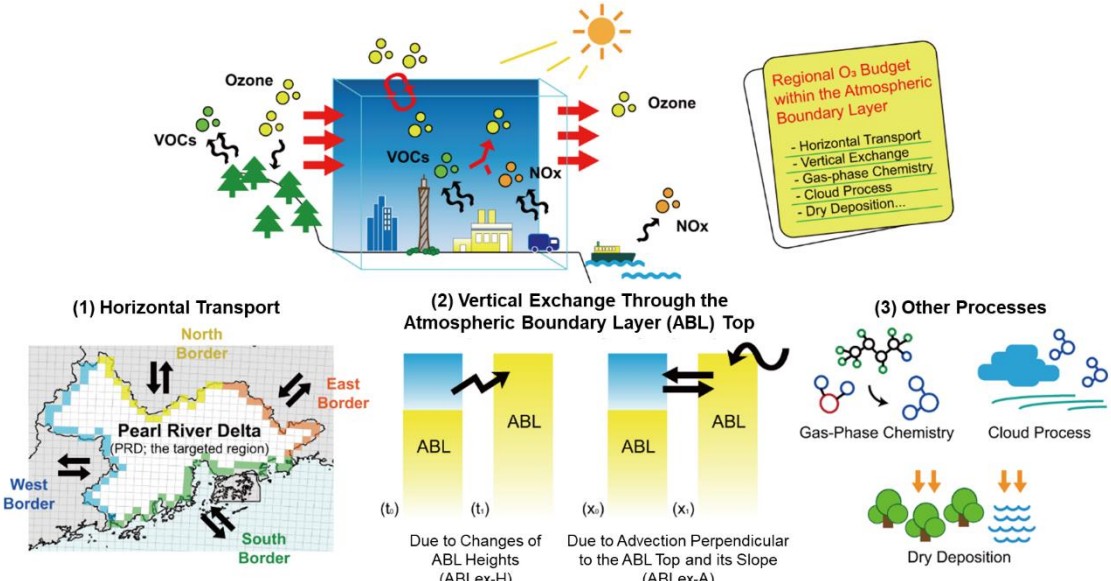


**Figure 1.** Schematic illustration of O$_3$ budgets (the upper panel) and O$_3$-related processes considered (the lower panel): (1) Horizontal transport through the north, south, west and east borders of the Pearl River Delta (PRD) (the distributions of the PRD grids are also shown: yellow, green, blue, orange for the north, south, west and east border grids, respectively, and white for the non-border grids); (2) Vertical exchange through the atmospheric boundary layer (ABL) top, including the process due to the changes of ABL heights (ABLex-H) and advection perpendicular to the ABL top and its slope (ABLex-A); (3) Other processes, including gas-phase chemistry, cloud process and dry deposition for this study.

168

Other processes in the O$_3$ budgets include gas-phase chemistry (including daytime photochemical O$_3$ production, O$_3$ titration by NO and O$_3$ depletion with unsaturated VOCs, etc.), cloud process (including below and in-cloud mixing, aqueous-phase chemistry, wet deposition; Liu et al., 2011) and dry deposition. The contributions of these processes are all calculated based on the output of the PA module in CMAQ. In a word, their contributions in the O$_3$ mass budget are obtained by summing up the contributions in all grid cells within the ABL of the PRD, and their contributions in the O$_3$ concentration budget are the corresponding contributions to O$_3$ mass divided by the volume of the ABL of the PRD. Since diffusion through the side and top boundaries of the region is expected to have a negligible influence on the variations of both O$_3$ concentration and mass, we did not consider this process in O$_3$ budget calculations.

177

The calculation process of the two O$_3$ budgets is summarized as follows. Based on multiple output files of WRF and CMAQ, firstly, the contributions of all considered O$_3$-related processes to O$_3$ mass changes and volumes/volume changes linked to these processes within the ABL are calculated in nearly all grids of the modelling domain. We developed the post-processing tool *flux_4d_cal* to conduct the above calculations. Afterwards, the regional-level O$_3$ mass and concentration budgets are quantified based on the results of the first-step calculations. Particularly, the method described in Sect. 2.3 is applied to

estimate the contributions of $O_3$-related processes in the $O_3$ concentration budget. More detailed descriptions of the
calculation process can be found in Text S1.

## 2.2 Transport contributions in the $O_3$ mass budget

The method by Yang et al. (2012) and Chang et al. (2018) was applied to quantify the contributions of horizontal transport in
the $O_3$ mass budget. For instance, the contribution of the advection through the west/east interface of a grid column within
the ABL to total $O_3$ mass ($F_{htrans}$) in the column during the time interval $dt$ is calculated as:

$$F_{htrans} = \int_0^H c_{O_3} uL \, dz \, dt \tag{4}$$

where $L$ is the width of the grid (equal to the horizontal resolution in the model); $dz$ is the height of vertical layers. For
advection through the north/south interface, the calculation is similar to Eq. (4), except for using v instead of u. $F_{htrans}$
values through all interfaces between the border grids and the outer region were calculated. Afterwards, they are summed up
separately according to the types of borders as the net contributions of horizontal transport through the north, south, west and
east borders of the PRD in the $O_3$ mass budget.

Following Sinclair et al. (2010) and Jin et al. (2021), the contribution of vertical exchange through the ABL top to $O_3$ mass
($F_{ABLex}$) during the time interval $dt$ can be expressed as:

$$F_{ABLex} = F_{ABLex-H} + F_{ABLex-A} = c_{O_3\_h} \frac{\partial H}{\partial t} L^2 dt + c_{O_3\_h} \left( u_h \frac{\partial H}{\partial x} + v_h \frac{\partial H}{\partial y} - w_h \right) L^2 dt \tag{5}$$

where $c_{O_3\_h}$ is the $O_3$ concentration at the ABL top. The two terms on the right-most side of Eq. (5) separately describe the
contributions of ABLex-H and ABLex-A (denoted separately as $F_{ABLex-H}$ and $F_{ABLex-A}$). $F_{ABLex}$ values in all the grids over
the PRD were summed up to derive the net contribution of vertical exchange through the ABL top in the $O_3$ mass budget.

## 2.3 Transport contributions in the $O_3$ concentration budget

It is difficult to directly apply Eq. (1) in the quantification of transport contributions in the regional-level $O_3$ concentration
budget. Therefore, a different approach was applied, which is introduced as follows.

Suppose that an air parcel with a total volume of $dV$ is transported into the ABL of the PRD (its original volume is $V$). The
variation of $\langle c_{O_3} \rangle$ under the influence of horizontal transport ($d\langle c_{O_3} \rangle_{htrans}$) can be written as:

$$d\langle c_{O_3} \rangle_{htrans} = \frac{F_{htrans} + \langle c_{O_3} \rangle (V - dV)}{V} - \langle c_{O_3} \rangle = \frac{F_{htrans} - \langle c_{O_3} \rangle \, dV}{V} \tag{6}$$

Since ABLex-A is also an advection process, its contribution in the $O_3$ concentration budget ($d\langle c_{O_3} \rangle_{ABLex-A}$) can be
quantified using a similar formula as Eq. (6), except for using $F_{ABLex-A}$ instead of $F_{htrans}$.

Through ABLex-H, air parcels in the residual layer and/or free atmosphere are merged into the ABL or vice versa. Thus, the
variation of $\langle c_{O_3} \rangle$ under its influence ($d\langle c_{O_3} \rangle_{ABLex-H}$) is expressed as:

$$d\langle c_{O_3} \rangle_{ABLex-H} = \frac{F_{ABLex-H} + \langle c_{O_3} \rangle V}{V + dV} - \langle c_{O_3} \rangle = \frac{F_{ABLex-H} - \langle c_{O_3} \rangle \, dV}{V + dV} \tag{7}$$


If the targeted region is small enough, the expressions of $d\langle c_{O_3} \rangle_{htrans}$ and $d\langle c_{O_3} \rangle_{ABLex-H}$ in Eqs. (6) and (7) can be
transformed to the corresponding terms in Eq. (1), confirming the applicability of the above calculations (for details, see
Text S2). All variables in Eqs. (6) and (7) can be quantified by the post-processing tool *flux_4d_cal*, making the method
feasible and suitable for the afterward calculations of the regional-scale $O_3$ concentration budget.

However, due to the prominent diurnal cycle of ABL, $V$ in Eqs. (6) and (7) may change notably within an hour, leading to
bias in the hourly estimations of $d\langle c_{O_3} \rangle_{htrans}$, $d\langle c_{O_3} \rangle_{ABLex-H}$ and $d\langle c_{O_3} \rangle_{ABLex-A}$ when using $V$ at the start and end of the
hour. This problem also applies to the calculation of contributions from other $O_3$-related processes. In order to reduce the
potential bias caused by the different selections of $V$, we designed two calculation paths for the hourly $O_3$ concentration
budget (Fig. S1):

- $O_3$ mass change $\rightarrow$ ABL volume change
- ABL volume change $\rightarrow$ $O_3$ mass change

where only $O_3$ mass or ABL volume changes in each calculation step. The contribution of ABLex-H to $O_3$ concentration can
be viewed as the net effects of ABL volume change and $O_3$ being transported into/out of the ABL: ABL volume change due
to ABL development (collapse) leads to lower (higher) $O_3$ concentration, and $O_3$ transported into (out of) the ABL through
ABLex-H leads to $O_3$ increase (decrease). These contributions are quantified separately in the ABL volume and $O_3$ mass
change step. The contributions of horizontal transport, ABLex-A and non-transport processes are quantified only in the $O_3$
mass change step. The contribution of each process to the variation of $O_3$ concentration is calculated using both paths, and
the mean value of two results serves as an estimation close to its real contribution in the $O_3$ concentration budget.

**2.4 Difference between the two $O_3$ budgets**

The difference between the two $O_3$ budgets is linked to the varied effect of transport on $O_3$ mass and concentration. Suppose
that the mean $O_3$ concentration in the transported air parcels is $\langle c_{O_3} \rangle_{trans}$. For horizontal transport, its contributions in the $O_3$
mass and concentration budgets can be separately written as:

$$F_{htrans} = \langle c_{O_3} \rangle_{trans} \, dV \tag{8}$$

$$d\langle c_{O_3} \rangle_{htrans} = \frac{dV}{V} \left( \langle c_{O_3} \rangle_{trans} - \langle c_{O_3} \rangle \right) \tag{9}$$

Apparently, $F_{htrans}$ is related to the $O_3$ concentrations in the transported air parcels, but not to those in the studied region. It
indicates the amount of $O_3$ mass transported into or out of the region. Whether it is positive or negative only depends on the
direction of transport — $O_3$ being transported into (out of) the region leads to the increase (decrease) of $O_3$ mass, which
corresponds to a positive (negative) contribution in the $O_3$ mass budget. In contrast, $d\langle c_{O_3}\rangle_{htrans}$ quantifies how much
horizontal transport alters regional-mean $O_3$ concentrations, and is linked to the difference between $O_3$ concentrations in the
transported air parcels and the studied region (Eq. (9)). $O_3$ being transported into (out of) the region does not necessarily
result in a higher (lower) $O_3$ concentration. For instance, when clean air parcels with relatively low $O_3$ levels are transported
into the region, they dilute $O_3$ pollution and reduce $O_3$ concentration ($d\langle c_{O_3}\rangle_{htrans} < 0$). Given that ABLex-A is also an
advection process, the above difference also applies to this process. For ABLex-H, its contributions in the $O_3$ mass and
concentration budgets are expressed as:

$$F_{ABLex-H} = \langle c_{O_3}\rangle_{trans}\, dV \qquad (10)$$

$$d\langle c_{O_3}\rangle_{ABLex-H} = \frac{dV}{V + dV}\left(\langle c_{O_3}\rangle_{trans} - \langle c_{O_3}\rangle\right) \qquad (11)$$

Similarly, ABL development and collapse lead to the increase and decrease of $O_3$ mass, respectively, but whether they
contribute to higher or lower $O_3$ concentration also depends on the difference between $O_3$ concentration in the transported air
parcels and that in the region. Based on the above discussion, these transport processes all show different effects on $O_3$ mass
and concentration — the effect of transport on the variations of $O_3$ mass is only related to the characteristics of the
transported air parcels, namely their volumes and $O_3$ concentrations within (Eqs. (8) and (10)), while how transport
contributes to the variations of $O_3$ concentration is linked to the difference between $O_3$ concentrations in the transported air
parcels and the region (Eqs. (9) and (11)).

To properly analyse the impact of transport and photochemistry on the regional origins of $O_3$, it is required to identify the
regional origins of the "new $O_3$" into the studied region and the "disappeared $O_3$" out of the studied region contributed by
various $O_3$-related processes, rather than how these processes lead to the variations of $O_3$ concentration. Thus, the influence
of transport and photochemistry on the results of $O_3$ source apportionment can be explored by the $O_3$ mass budget, but not by
the $O_3$ concentration budget. By utilizing the BFM source apportionment method in combination with the $O_3$ mass budget
calculation, we can identify the regional origins of $O_3$ mass increase and decrease due to transport and photochemistry, and
explain how these processes determine the results of $O_3$ source apportionment in the PRD.
**2.5 Model setup and validation**
The $O_3$ concentration and mass budgets within the ABL of the PRD were calculated based on the WRF-CMAQ modelling
results by Qu et al. (2021a). The WRF (version 3.2) and CMAQ (version 5.0.2) models were used to simulate the
meteorological and pollutant fields, respectively. Two domains with the resolution of 36 and 12 km (denoted as d01 and d02
hereafter) were set up for the one-way nested simulations, and the results in the finer d02, which includes the PRD and most
areas in East and Central China (Fig. 2), were used in the calculations of $O_3$ budgets. To represent the contributions of global
background to $O_3$, the initial and boundary conditions for the coarse d01 domain were provided from the global model, the
Model for Ozone and Related Chemical Tracers, version 4 (MOZART-4). The PRD inventory provided by the Guangdong
Environmental Monitoring Centre, the Multi-resolution Emission Inventory for China (MEIC) inventory for the mainland
China (He, 2012), the MIX inventory for the Asian regions outside of mainland China (Li et al., 2017) and biogenic
emissions simulated by the Model of Emissions of Gases and Aerosols from Nature (MEGAN; version 2.10) model were
used in the simulations. SAPRC07 (Carter, 2010) and AERO6 were applied as the gas-phase chemistry mechanism and the
aerosol scheme, respectively. The simulations of $O_3$ pollution in the PRD were performed for October 2015 (October 11–
November 10, 2015) and July 2016 (July 1–31, 2016), which serve as the representative months in autumn and summer,
respectively. Here, $O_3$ polluted days are defined when the maximum hourly $O_3$ concentrations of the day exceed 200 μg/m$^3$,
or the maximum 8-hour average $O_3$ concentrations of the day exceed 160 μg/m$^3$ (both are the Grade-II $O_3$ thresholds in the
Chinese National Ambient Air Quality Standard) in any municipality of the PRD. According to this definition, there were 16
and 12 $O_3$ polluted days in the two months, respectively (more information is given in Table S1). The mean $O_3$ budgets
during these $O_3$ polluted days of two seasons were separately calculated and discussed in the present study.

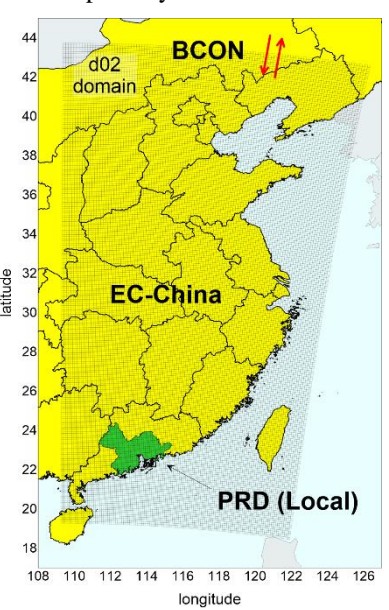


**Figure 2.** The spatial distributions of the d02 modelling domain and source regions. The d02 domain is displayed as the nested areas in the
figure. PRD, Pearl River Delta; EC-China, East and Central China; BCON, the boundary conditions of d02 modelling, or the contributions
of sources outside the d02 domain.

We evaluated the performance of WRF-CMAQ modelling based on multiple observational datasets. The modelling results of
meteorological parameters (including temperature, relative humidity and wind speed), $O_3$, $NO_2$ concentrations and the
mixing ratios of hydrocarbons were validated with corresponding observations in the PRD by Qu et al. (2021a). The
performance of the model in simulating the above variables was overall satisfying with low biases and high correlations (for
details, see Qu et al., 2021a). In this study, we further compared the modelled ABL height, the vertical profiles of wind
speed, direction and $O_3$ mixing ratio in Hong Kong (located in the south PRD) with corresponding observations from the
IAGOS (In-service Aircraft for a Global Observing System; Petzold et al., 2015) dataset. The modelled ABL heights showed
similar hourly variations during the day as the observational results (R = 0.76), with a mean bias of -1.1 m (Fig. S2). The
mean biases of mean wind speeds are within the range of $\pm$ 1 m/s in all considered height ranges (0-1 km, 1-2 km, 2-5 km),
and the results of the IAGOS and WRF model indicate similar variations of prevailing wind directions in different seasons
and height ranges (Fig. S3). Moreover, modelled $O_3$ mixing ratios in Oct. 2015 are overestimated by 6% and 26% in the
height range of 0-1 km and 1-2 km, respectively, and sufficiently illustrate the development, maintenance and dissipation of
$O_3$ pollution during the month (Fig. S4). More detailed evaluations on the model performance of these parameters are
presented in Text S3 of the Supplement. Overall, the model performance is acceptable, indicating that the model can provide
reasonable data for the calculations of $O_3$ budgets.

If the calculation methods and assumptions are reasonable, the conservation of $O_3$ concentration and mass budgets, described
as

$$\frac{\partial \langle c_{O_3} \rangle (or\ m_{O_3})}{\partial t} - \left( S_{htrans} + S_{ABLex} + S_{chem} + S_{cloud} + S_{ddep} \right) = 0 \tag{12}$$

can be achieved (the terms $S_{htrans}, S_{ABLex}, S_{chem}, S_{cloud}$ and $S_{ddep}$ indicate the contributions of horizontal transport, vertical
exchange through the ABL top, gas-phase chemistry, cloud process and dry deposition, respectively, in the $O_3$ concentration
or mass budgets). Therefore, we used Eq. (12) to examine the validity of $O_3$ budget calculations. Total $O_3$ masses at the start
and end of each hour were directly used to calculate the hourly variations of $O_3$ mass ($\frac{\partial m_{O_3}}{\partial t}$). Besides these two parameters,
the volumes of the ABL of the PRD at the start and end of each corresponding hour (calculated using ABL heights in all the
PRD grids) are also needed to calculate the hourly variations of $O_3$ concentration ($\frac{\partial \langle c_{O_3} \rangle}{\partial t}$). The contributions of various $O_3$-
related processes in the $O_3$ concentration and mass budgets were quantified using the method introduced in Sect. 2.1-2.3. As
displayed in Fig. 3, hourly variations of $O_3$ concentration/mass and the corresponding net contributions from all $O_3$-related
processes show good correlations ($R^2 > 0.9$), with all fitted lines close to the 1:1 line. Thus, the conservation is overall met
for the two $O_3$ budgets in both representative months, allowing for further analyses based on the quantified budgets.
**2.6 Identifying regional origins of $O_3$ mass changes due to transport and photochemistry**
The question to be addressed is how $O_3$-related processes determine the regional origins of $O_3$. By combining the $O_3$ mass
budget calculations with the BFM source apportionment method, we identified the regional origins of $O_3$ mass changes due
to transport and photochemistry (gas-phase chemistry). Here, the interest lies in the contributions of emissions in the PRD
(also defined as local emissions), in other regions within d02 (mainly East and Central China, hereafter denoted as EC-
China), and in regions outside the d02 (the boundary conditions (BCON) of d02 modelling; representative of the background
sources). The distribution of these source regions is shown in Fig. 2. Besides the base scenario where all emissions in d02
were considered in simulations, three sensitivity scenarios were additionally simulated:
•   The PRD_zero scenario: All emissions (including anthropogenic and biogenic emissions; the same below) in the

321        PRD were zeroed out;

•   The EC-China_zero scenario: All emissions in the EC-China were zeroed out;
•   The All_zero scenario: All emissions within d02 were shut down.

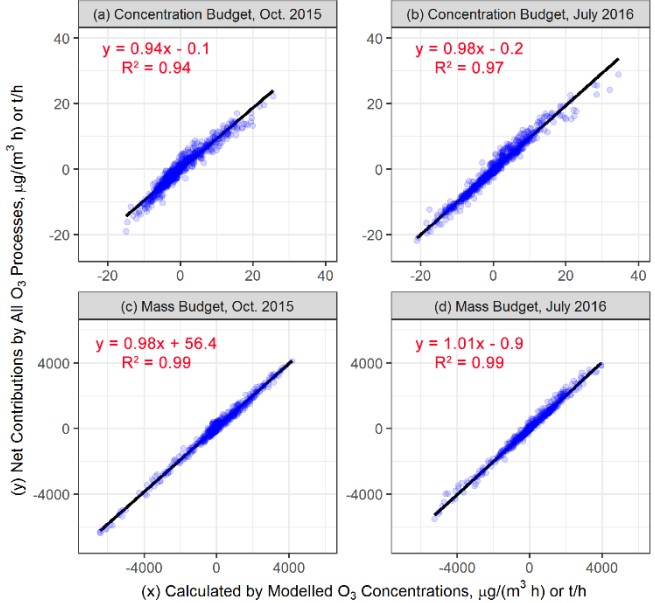


**Figure 3.** The examinations of $O_3$ budget conservation in Oct. 2015 (a,c) and July 2016 (b,d) for the hourly $O_3$ concentration budget (a-b)
and mass budget (c-d). The units for the $O_3$ concentration and mass budgets are $\mu g/(m^3\ h)$ and $t/h$, respectively. The solid black lines in the
plots are the fitted lines.
The hourly contributions of the process $i$ in the $O_3$ mass budget were quantified using the same method outlined in Sect. 2.1-
2.2 for the base scenario and three sensitivity scenarios, denoted as $f_{i,base}$, $f_{i,PRD\_zero}$, $f_{i,EC-China\_zero}$, and $f_{i,all\_zero}$,
respectively. These parameters enable the determination of the contributions of emissions from the PRD and EC-China as
well as the background sources (BCON) to the $O_3$ mass increase and decrease due to various $O_3$-related processes. The
contributions of BCON in the $O_3$ mass changes due to the process $i$ ($F_{i,BCON}$) can be estimated directly as the contributions of
the process $i$ to the $O_3$ mass in the All_zero scenario:

$$F_{i,BCON} = f_{i,all\_zero} \tag{13}$$

For the contributions of the PRD and EC-China emissions from the process $i$ (separately denoted as $F_{i,PRD}$ and $F_{i,EC-China}$),
they can be derived in two ways: 1) by subtracting simulations with zeroed studied emissions from the base case simulation
(top-down BFM); 2) by subtracting simulations without all emissions from simulations accounting only for studied
emissions (bottom-up BFM). Due to the non-linear response of $O_3$ to precursor emissions, the results from top-down and
bottom-up BFM can differ, which may lead to the non-additivity of the results (the sum of all contributions is not equal to
the concerned metric; here, $F_{i,PRD} + F_{i,EC-China} + F_{i,BCON} \neq f_{i,base}$). Therefore, we estimated $F_{i,PRD}$ and $F_{i,EC-China}$ as the
average values of the contributions by using top-down BFM and bottom-up BFM:

$$F_{i,PRD} = \frac{1}{2}[(f_{i,base} - f_{i,PRD\_zero}) + (f_{i,EC-China\_zero} - f_{i,all\_zero})] \tag{14}$$

$$F_{i,EC-China} = \frac{1}{2}[(f_{i,base} - f_{i,EC-China\_zero}) + (f_{i,PRD\_zero} - f_{i,all\_zero})] \tag{15}$$

It should be noted that to identify the origins of both "new $O_3$" into the region and "disappeared $O_3$" out of the region, the
positive and negative contributions of $O_3$-related processes to the $O_3$ mass in the PRD grids were separately summed up for
the base and sensitivity scenarios and quantified using Eqs. (13-15).

## 3 Analyses and comparisons of $O_3$ concentration and mass budget

### 3.1 $O_3$ concentration budget

The upper panels of Fig. 4 show the mean diurnal changes of the $O_3$ concentration budget within the ABL of the PRD.
According to the net contributions from all $O_3$-related processes considered, ABL-mean $O_3$ concentration increased during
most hours in the daytime, with the highest rates occurring in the early morning (8:00-10:00 local time (LT) in autumn, 7:00-
9:00 LT in summer). The reduction of ABL-mean $O_3$ concentration in the late afternoon and at night was also considerable.
Its rate reached the maximum value near the sunset time (~18:00 LT in autumn, ~19:00 LT in summer) and gradually
decreased throughout the night. The following question is then raised on the suitability of the budget targeting on ABL-mean
$O_3$ concentration to explain the variations of $O_3$ concentrations near the ground. To answer this question, we compared the
hourly changes of modelled ABL-mean $O_3$ concentration with those of observed and modelled mean near-surface $O_3$
concentrations in 18 sites of the Guangdong-Hong Kong-Macao PRD Regional Air Quality Monitoring Network
(distributions shown in Fig. S5). As presented in Fig. S6, these datasets display similar patterns of $O_3$ diurnal changes. Since
$O_3$ was well mixed within the ABL (Fig. S4), especially during daytime when $O_3$ levels are higher than those at night, the
budget of ABL-mean $O_3$ concentration can reveal the influences of transport and photochemistry on the variations of overall
$O_3$ levels as well as the causes of $O_3$ pollution in the targeted region.

Next, the contributions of various $O_3$-related processes in the $O_3$ concentration budget are discussed as follows:
•   Gas-phase chemistry: Figure 4 shows that gas-phase chemistry controlled almost exclusively the $O_3$ concentration

budget. During the morning hours, which are defined as the period from sunrise (~6:00 LT in autumn, ~5:00 LT in

summer) to the $O_3$-peak hour (~14:00 LT), gas-phase chemistry (photochemistry) contributed to, on average, 74%

and 95% of the $O_3$ concentration increase in autumn and summer, respectively. These contributions are notably

higher than the contributions of transport in the same periods (25% in autumn, 5% in summer). In the afternoon,

gas-phase chemistry was still the main process to maintain high $O_3$ concentrations within the PRD, but its

contributions gradually decreased. However, this process led to decreased $O_3$ concentration at night, suggesting the

impact of $O_3$ titration by emitted NO and $O_3$ depletion with unsaturated VOCs. It may also be related to the

production of particle nitrate through $N_2O_5$ hydrolysis (Qu et al., 2021b).

•   Transport: The dominance of gas-phase chemistry in the $O_3$ concentration budget does not mean that the influence

of transport on $O_3$ concentration can be neglected all day long. Considerable contributions of transport (mainly by

ABLex-H) to $O_3$ concentration increase are found during 2-3 hours after sunrise, with the highest hourly mean

contributions reaching ~40% and ~25% in autumn and summer, respectively. This result indicates the notable

influence of air masses with high $O_3$ concentrations being entrained from residual layers on near-surface $O_3$

pollution. ABLex-A and horizontal transport may contribute to the increase or decrease of ABL-mean $O_3$

concentration, depending on the $O_3$ levels in air parcels transported into and out of the region (further analysis is

provided in Sect. 3.3). Overall, these two transport processes had only limited contributions to the variations of $O_3$

concentration.

•   Other processes: Dry deposition contributed to a considerable decrease in $O_3$ concentration, especially during

daytime, and thus served as an important sink process for near-surface $O_3$. Besides, cloud process was also an

important sink process for $O_3$ in summer, which might be related to the convective vertical transport of $O_3$.


In summary, the results of the $O_3$ concentration budget indicate that gas-phase chemistry played a major role in the variations
of $O_3$ concentrations in the PRD. In particular, photochemistry led to the rapid formation of $O_3$ pollution during daytime,
rather than transport. Our conclusions agree well with those in earlier studies on the $O_3$ concentration budget (Lenschow et
al., 1981; Hou et al., 2014; Trousdell et al., 2016; Su et al., 2018; Tan et al., 2018; Tan et al., 2019; Trousdell et al., 2019; Yu
et al., 2020; Li et al., 2021; Yan et al., 2021).

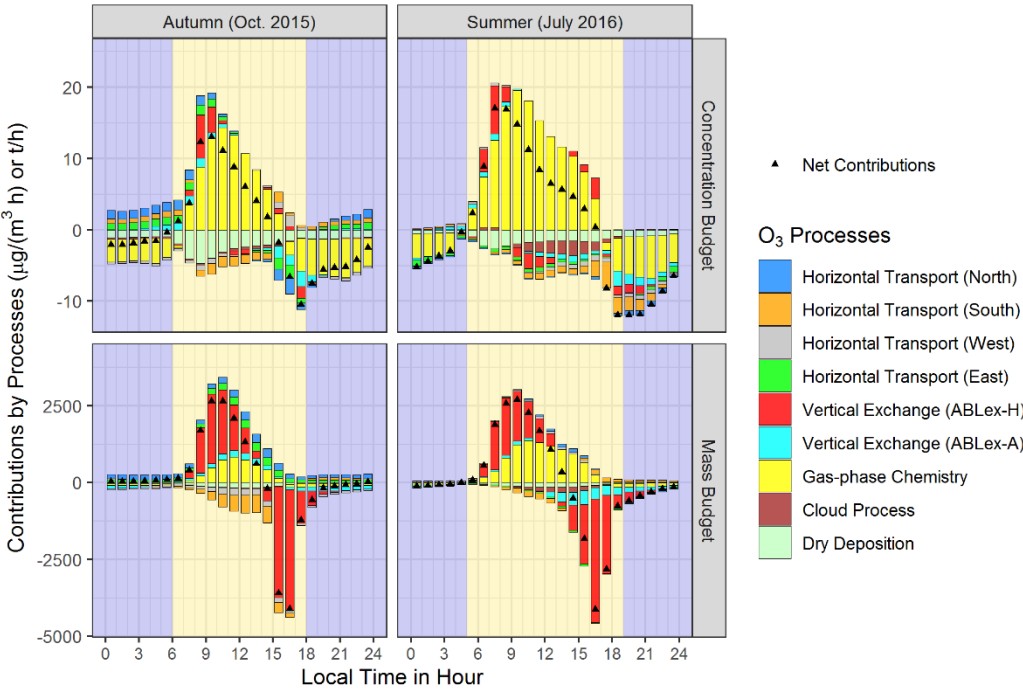


**Figure 4.** Mean diurnal changes of the $O_3$ concentration budget (upper panels) and mass budget (lower panels) on the polluted days of
representative months in autumn (Oct. 2015; left panels) and summer (July 2016; right panels) within the atmospheric boundary layer of
the Pearl River Delta. The units for the $O_3$ concentration and mass budgets are $\mu g/(m^3\ h)$ and t/h, respectively. Backgrounds in yellow and
dark blue indicate daytime and nighttime periods, respectively.

**3.2 $O_3$ mass budget**

The results of the $O_3$ mass budget are displayed in the lower panels of Fig. 4. The total $O_3$ mass within the ABL of the PRD
increased during the morning hours, decreased rapidly in the afternoon and slowly at the early night, then remained stable
until sunrise in both seasons. The change of total $O_3$ mass agrees well with the ABL diurnal cycle (Lee, 2018) — daytime
ABL development (or collapse) and notable $O_3$ mass increase (or decrease) almost occurred simultaneously, and the
negligible changes in $O_3$ mass during most hours of the night may be linked to the small variations of stable ABL.

We analysed the contributions of various $O_3$-related processes in the $O_3$ mass budget as well, presented as follows:

- Transport: Unlike the results of the $O_3$ concentration budget, transport plays a prominent role in the $O_3$ mass budget.
  On average, it contributed 78% and 53% to $O_3$ mass increase during the morning hours of autumn and summer,
  respectively, and over 90% to $O_3$ mass decrease during the afternoon hours of both seasons (14:00-18:00 LT in
  autumn and 14:00-19:00 LT in summer). Most $O_3$ was transported into or out of the PRD by vertical exchange
  through the ABL top, especially ABLex-H, which links the diurnal changes of $O_3$ mass and ABL. That is to say,
  when the height of ABL rise (drop) rapidly, a big amount of $O_3$ is transported into (out of) the ABL through the
  ABLex-H. The contributions of ABLex-A and horizontal transport to $O_3$ mass change were relatively limited.

However, they indicate well the characteristics and variations of regional wind fields in the PRD (more details are
provided in the next section).
• Gas-phase chemistry: Gas-phase chemistry (photochemistry) also contributed to the increasing $O_3$ mass in the
daytime, especially in summer. However, its mean contributions during the morning hours (22% in autumn, 47% in
summer) were lower than those of transport.
• Other processes: Dry deposition and cloud process both acted as $O_3$ sink processes, but with negligible
contributions to $O_3$ mass.

Based on the above discussions, transport tends to be more important than photochemistry in the $O_3$ mass budget, which
differs from the conclusions of the $O_3$ concentration budget. The main role of transport, especially ABLex-H, in the $O_3$ mass
budget suggests the marked impacts of the ABL diurnal cycle on regional $O_3$ pollution. Despite of less notable influence of
transport on $O_3$ concentration increase in comparison to that of photochemistry, massive $O_3$ being transported into the ABL
of the targeted region during the morning hours nearly determines the regional origins of $O_3$ pollution. Quantified results
combining the $O_3$ mass budget and source apportionment are further discussed in Sect. 4.
**3.3 Influences of regional wind fields on $O_3$ pollution: more analyses of transport contributions in $O_3$ budgets**
As discussed before, the contributions of horizontal transport and ABLex-A were relatively limited in the two $O_3$ budgets.
However, they illustrate well the influences of regional wind fields, including the seasonal prevailing winds and local
circulations (sea breezes), on $O_3$ pollution in the PRD. Two main findings from the analyses of these transport contributions
are presented below.
**3.3.1 Transport contributions in autumn: The characteristics of prevailing winds**
In the PRD, northerly and easterly winds prevail in autumn (as indicated by the wind roses in Fig. S3). Correspondingly, $O_3$
was transported into the PRD through its north and east borders, out of the PRD through the south and west borders, as
indicated by the $O_3$ mass budget (Fig. 4). $O_3$ masses transported out of the PRD were generally higher than those transported
into the PRD during daytime. This is attributed to higher $O_3$ concentrations in the downwind regions due to $O_3$ production
mostly from local emissions. "Low $O_3$ in, high $O_3$ out" also explains why horizontal transport led to the net decrease of $O_3$
concentration during daytime. At night, $O_3$ was still transported into the region through the north and east borders of the
PRD, but these processes contributed to the increase of $O_3$ concentrations. That is to say, with relatively higher $O_3$
concentrations compared to those in the $NO_x$-titrated urban atmosphere, air parcels transported from the upwind outskirts
served as the supply to slowdown night-time $O_3$ level decrease in the PRD due to chemistry and deposition.

The daytime contributions of ABLex-A in the $O_3$ mass budget also indicate the effects of prevailing northerly winds. The
PRD has mountainous regions in the northern, western and eastern outskirts, as well as urban regions with lower altitudes in
the central plain (Fig. S5). As shown in Fig. S7a-b, the positive contributions of ABLex-A through the ABL top (in the z-
direction) can be found in the mountainous northern PRD, suggesting that northerly winds resulted in the downward
transport of $O_3$ along the terrain. Daytime ABL heights in urban regions were, in general, higher than those in the
surrounding mountainous regions, which is the other reason why $O_3$ can be transported through the ABL slope (in the x-/y-
direction) near the urban-rural interfaces when northerly wind prevailed (Fig. S7c-d). For the $O_3$ concentration budget,
ABLex-A contributed to increased $O_3$ concentration during several hours after sunrise but decreased $O_3$ concentration in the
afternoon. This different effect is attributed to different comparison results between ABL and above-ABL mean $O_3$
concentrations in the two periods ($O_3$ concentration above the ABL is overall higher than that within the ABL in the
morning, while the opposite is for the afternoon; Fig. S4).

**3.3.2 Transport contributions in summer: The influence of sea breezes**
Although southerly winds normally prevail in summer in the PRD (Fig. S3), on $O_3$ polluted days, air parcels from other
directions could also influence the region (Qu et al., 2021a). Thus, the mean contribution of horizontal transport to $O_3$ mass
in summer was lower than in autumn. Of particular interest is the variation of the contributions of horizontal transport
through the south border of the PRD before and after ~14:00 LT, as indicated by the results of the $O_3$ mass budget (Fig. 4).
Besides, both $O_3$ budgets suggest notable $O_3$ mass and concentration decreases due to ABLex-A in the afternoon. These
phenomena are both related to the influence of sea breezes.

Figure 5 shows the near-surface wind roses at 14:00, 16:00 and 18:00 LT of $O_3$ polluted days in July 2016 based on the
observational and modelling results in the national meteorological sites within the PRD. At 14:00 LT, the main wind
directions were W, SW and NW in both datasets. More S and SE winds occurred in later hours, and they became the
prevailing winds at 18:00 LT, suggesting the gradual development of sea breezes in the PRD. Thus, $O_3$ was originally
transported out of the PRD through the south border with negative contributions to $O_3$ mass; in the late afternoon, sea
breezes reversed the directions of $O_3$ transport, resulting in positive contributions to $O_3$ mass by horizontal transport through
the south border (Fig. 4). Moreover, the development of sea breezes is connected to the changes of wind fields not only
horizontally, but also vertically. Taking the $O_3$ polluted day July 24th, 2016 for example, the cross-section of $O_3$
concentrations and wind fields in the PRD at 16:00 LT of the day is shown in Fig. 6 (the cross-section is made along the
113.2° E longitude, ranging from 26.0° to 20.0° N in latitude). Strong southerly wind and lower $O_3$ concentrations are found
in the southern PRD, indicating the influence of sea breezes during that time. Near the interfaces where sea breezes
encountered local air parcels (indicated by the drastic increase in $O_3$ concentrations from less than 100 μg/m$^3$ to about 100-
150 μg/m$^3$), updrafts occurred, suggesting the formation of sea breeze front (Ding et al., 2004; You and Fung, 2019). The
front promoted the upward transport of $O_3$ from the ABL, or considerable $O_3$ mass decrease due to ABLex-A. Both
horizontal transport and ABLex-A led to decreased $O_3$ concentrations, because under the effect of sea breezes, clean air
parcels were transported into the region and polluted air parcels were transported out of the region. The influence of sea
breezes can also be seen in autumn but was weaker and occurred later than in summer. Besides, in autumn, horizontal
transport through the south border of the PRD contributed to the increase of $O_3$ concentration at night, indicating the effects
of $O_3$ recirculation from the "$O_3$ pool" in the bay areas to the south of the PRD (Zeren et al., 2019; Zeren et al., 2022).

Through the calculations and analyses of transport contributions in the two $O_3$ budgets, the influences of complex transport
processes on multiple scales to $O_3$ concentration and mass can be well identified. These results provide a deeper
understanding of how transport influences regional $O_3$ pollution in the PRD.

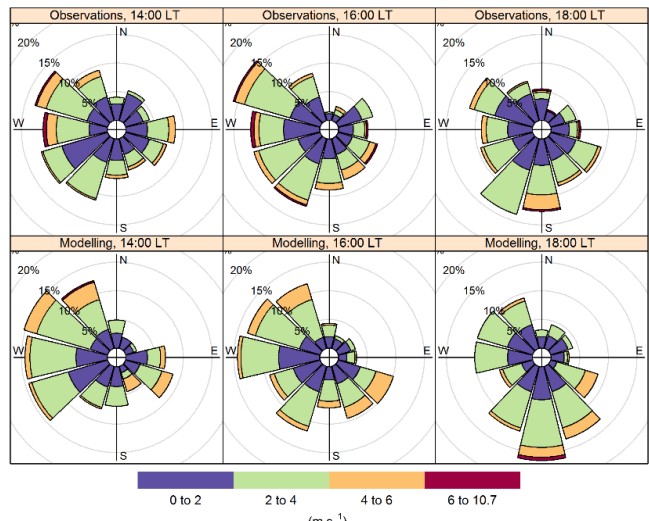


**Figure 5.** Wind roses at 14:00, 16:00, and 18:00 local time (LT) of the $O_3$ polluted days in July 2016 in the Pearl River Delta (PRD).
Observational and modelling wind speeds and directions in 29 national meteorological sites within the PRD were used for this figure.

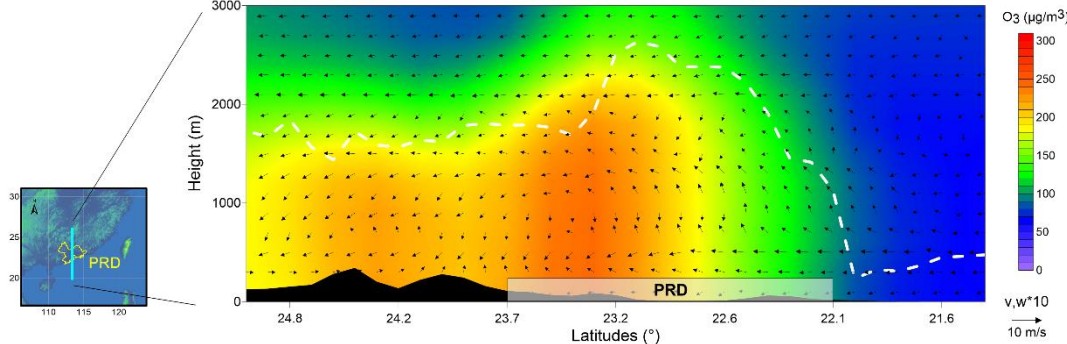


**Figure 6.** Cross-section of $O_3$ concentrations ($\mu g/m^3$) and wind fields at 16:00 local time on July 24th, 2016. The dashed white line
indicates the top of the atmospheric boundary layer. PRD, Pearl River Delta.

 **4 Effects of transport and photochemistry on the regional origins of O₃**

Based on reported publications (Li et al., 2012; Li et al., 2013; Yang et al., 2019; Gao et al., 2020), O₃ in the PRD is mostly
derived from emissions outside the PRD and background O₃, rather than local emissions. This is the same for the O₃ polluted
days in the representative months of autumn and summer in this study, when the contributions of non-local sources account
for, on average, 89% and 65% of the O₃ in the PRD, respectively, in 9:00-17:00 LT (55% and 32% contributed by BCON,
34% and 33% contributed by EC-China in the two months; Qu et al., 2021a). To explain why non-local sources are dominant
for O₃ in the PRD, by combining O₃ mass budget calculation with O₃ source apportionment (method introduced in Sect. 2.6),
we identified the regional origins of O₃ mass changes due to vertical exchange through the ABL top, horizontal transport and
gas-phase chemistry (Fig. 7). Here, the contributions of three sources to the O₃ mass increase and decrease were both
quantified. But further analyses focus on the results related to O₃ mass increase, because the origins of O₃ in the region are
more likely to be influenced by the "new O₃" transported into and produced within the PRD.

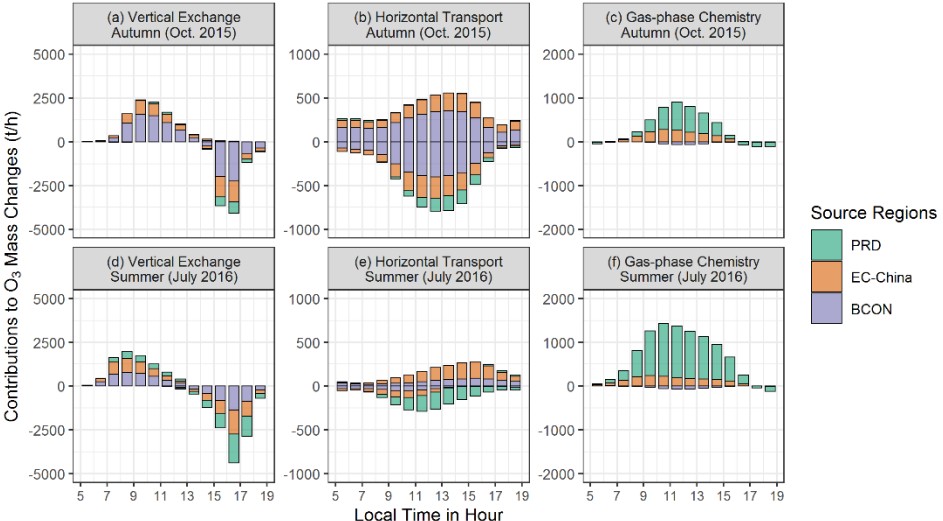


**Figure 7.** The regional origins of hourly O₃ mass changes contributed by (a,d) vertical exchange through the ABL top, (b,e) horizontal
transport, and (c,f) gas-phase chemistry on the polluted days of representative months in autumn (Oct. 2015; a-c) and summer (July 2016;
d-f). The results for the time window 5:00-19:00 LT are shown here. PRD, Pearl River Delta; EC-China, East and Central China; BCON,
the boundary conditions of d02 modelling, or the contribution of sources outside the d02. Note that the scales are different among the three
columns.

Through vertical exchange through the ABL top, massive non-local O₃ entered into the ABL of the PRD. In the morning-
hour O₃ mass increase due to this process, BCON and EC-China accounted for 65% and 31%, respectively, in autumn. By
contrast, local emissions only contributed 4% to this transported O₃ during the same period, suggesting that local O₃ was less
likely to be recirculated back to the PRD during daytime. In summer, the contribution of local emissions in the O₃ mass
transported into the region through vertical exchange was higher than in autumn, reaching 20% during the morning hours.
However, non-local sources still dominated the $O_3$ mass increase due to vertical exchange — the morning-hour contributions
in percentage of BCON and EC-China were 42% and 38%, respectively.

$O_3$ mass increase due to horizontal transport was connected to the contribution of non-local sources as well. In both seasons,
$O_3$ transported into the PRD originated almost exclusively from EC-China and BCON.

It is not surprising that most $O_3$ produced through photochemistry (daytime gas-phase chemistry) was related to local
emissions, of which the contributions accounted for 66% and 82% during the daytime of autumn (6:00-18:00 LT) and
summer (5:00-19:00 LT), respectively. The contributions of EC-China emissions in the daytime $O_3$ mass increase reached
34% and 18% in the two seasons, respectively, indicating that the influences of non-local $O_3$ precursor import on local $O_3$
photochemistry are also considerable in the PRD.

With the results of the $O_3$ mass budget and the regional origins of $O_3$ mass increase due to transport and photochemistry, the
effect of $O_3$-related processes on the origins of $O_3$ can be revealed. Based on the $O_3$ mass budget, the accumulated morning-
hour $O_3$ mass increase exceeded 10000 tons for both seasons, which is 6-9 times larger than the original $O_3$ mass in the ABL
of the PRD before sunrise (< 1500 tons). Thus, in the daytime, most $O_3$ in the PRD was the "new $O_3$" contributed by
transport and photochemistry, and the origins of $O_3$ within the region were nearly determined by these of newly transported
and produced $O_3$. By combining the $O_3$ mass budget and $O_3$ source apportionment, we identified the $O_3$ mass increase due to
$O_3$-related processes as local (PRD) and non-local (EC-China and BCON) contributions. According to the results discussed
before, high contributions of transport in the morning-hour $O_3$ mass increase and the dominance of non-local source
contributions in this part of new $O_3$ ensure that non-local sources contributed to most $O_3$ in the PRD. Moreover, differences
in the contributions of $O_3$-related processes in the $O_3$ mass budget as well as the origins of morning-hour $O_3$ mass increase
lead to varied origins of $O_3$ in the region. For instance, when comparing the results of $O_3$ source apportionment in the two
seasons, we found that the contributions of non-local sources (local emissions) to $O_3$ were lower (higher) in summer than in
autumn. It can be attributed to the combined effects of increased photochemistry contributions (or decreased transport
contributions) in the $O_3$ mass increase and reduced non-local source contributions in both transported and chemically
produced $O_3$ in summer. Collectively, these changes lead to reduced non-local contributions (or higher local contributions) to
$O_3$.

By influencing $O_3$ mass increase and its regional origins, transport and photochemistry determine the results of $O_3$ source
apportionment within the region. Specifically, transport (mainly ABLex-H) brings massive non-local $O_3$ into the region in
the morning, explaining why most $O_3$ in the PRD is derived from non-local sources. However, accompanied with the
simultaneous rapid increase of ABL volumes, this process has a relatively limited contribution to $O_3$ concentration increase
in comparison to photochemistry. The $O_3$ concentration budget only concerns the influence of $O_3$-related processes on the
variations of $O_3$ concentration, thus it fails to illustrate the effect of transport on the regional origin of $O_3$. Our results
highlight the difference between the $O_3$ concentration and mass budgets, which may result in distinct understandings about
the role of transport and photochemistry in regional $O_3$ pollution. To completely illustrate the effects of two $O_3$-related
processes on regional $O_3$ pollution, insights from both $O_3$ budgets are required.
**5 Conclusion and outlook**
To effectively alleviate $O_3$ pollution, it is important to understand the respective role of transport and photochemistry in
regional $O_3$ pollution. The $O_3$ concentration budget is widely used to quantify the contributions of these $O_3$-related processes
to the variations of $O_3$ concentrations, and it often concludes that photochemistry is the main contributor to the aggravation
of $O_3$ pollution. However, it does not explain why most of the $O_3$ is transported from the outside regions as indicated by $O_3$
source apportionment studies. To comprehensively illustrate the effects of transport and photochemistry on regional $O_3$
pollution, based on the modelling results of WRF-CMAQ, this study presents a method to quantify not only the $O_3$
concentration budget, but also the $O_3$ mass budget, in which the contributions of $O_3$-related processes (including transport
and photochemistry) to the variations of mean $O_3$ concentrations and total $O_3$ mass within the ABL of the PRD are separately
identified. The different effects of transport on $O_3$ concentration and mass were considered in the above calculations. The $O_3$
concentration budget in the PRD reveals that gas-phase chemistry, including daytime photochemistry and night-time $O_3$
titration/depletion, drives the variations of $O_3$ concentration. Particularly, photochemistry contributed 74% and 95% to the
$O_3$ concentration increase in the morning hours of autumn and summer months, respectively. In contrast, transport,
especially the vertical exchange through the ABL top, is the main process contributing to the $O_3$ mass increase in the
morning (78% and 53% in autumn and summer, respectively) and decrease in the afternoon (> 90%). The diurnal changes of
transport contributions in the two $O_3$ budgets are closely connected to the variations of the ABL and regional wind fields,
including the seasonal prevailing winds and local circulations (sea breezes), in the PRD. Massive $O_3$, mostly derived from
non-local sources, being transported into the ABL in the morning has a relatively limited influence on the $O_3$ concentration
increase (25% and 5% in autumn and summer, respectively) compared to photochemistry because of the rapid change of
ABL volumes at the same time. However, this process nearly determines the dominance of non-local source contributions
for daytime $O_3$ in the PRD. The two $O_3$ budgets show notable differences, but together they provide a more complete
overview of the effects of transport and photochemistry on regional $O_3$ pollution.

It should be noted that the conclusions in this study apply not only to $O_3$, but also to other pollutants with moderately long
atmospheric lifetimes, including fine particulate matter and some of its components. In theory, transport and chemical
transformations are both important processes for these pollutants. However, transport has different effects on the
concentration and mass of pollutants on an hourly scale, which is similar to the discussion in Sect. 2.4. Furthermore, besides
regional origins, the difference between the two budgets may also contribute to the inconsistency of other characteristics of
pollutants, such as the contributions of different reaction pathways and sensitivities to precursor emissions, identified by the
concentration budget and mass-based methods. When large quantities of pollutants with different characteristics are
transported into the region, the variation of their concentrations is often not perceptible and thus neglected in the
concentration budget. However, as indicated by this study, the transport processes are likely to change or even determine the
characteristics of pollutants within the region. Therefore, we suggest that attention should be paid to selecting a proper
budget type and using correct budget calculation methods in related research. But to fully reveal the effects of transport,
chemistry and other related processes on regional pollution, insights from both concentration and mass budgets are
necessary.

Uncertainty remains in the calculated $O_3$ budgets, which is partly related to the biases in the modelling results. Therefore,
supporting observations are essential for future research. Recent progress in observational techniques (Zhao et al., 2021;
Zhou et al., 2021) has enabled three-dimensional measurements of meteorological parameters and $O_3$ concentrations with
high spatiotemporal resolution and coverage. These data can be used not only for the model validation of key parameters in
budget calculations, but also for the comparisons between observation- and modelling-based contributions by various $O_3$-
related processes in $O_3$ budgets (Kaser et al., 2017). The comparison of contributions by $O_3$-related processes is indicative of
the main uncertainties in $O_3$ pollution modelling, and is therefore also important for further model developments.

The present study concluded that transport and gas-phase chemistry play the main role in the $O_3$ mass and concentration
budgets, respectively. As a consequence of our assessment, the following is suggested for policy-makers. For areas where
non-local emissions notably contribute to $O_3$, emission reduction in the upwind regions can reduce the overall $O_3$
concentrations effectively, which is a crucial step towards the long-term improvement of regional air quality. However, for
short-term air pollution control, this strategy is not efficient because emission reduction in upwind regions may need to start
days earlier before the polluted periods. In contrast, reducing local emissions is expected to lower the rapid daytime $O_3$
concentration increase efficiently and, thereby, $O_3$ peak levels in the short term, as highlighted by the $O_3$ concentration
budget. The choice of the better strategy to be applied should depend on the specific objectives of $O_3$ control (mean levels vs.
peak levels; long-term vs. short-term), which are set based on a more in-depth understanding of $O_3$ effects on human health,
crop yields and ecosystems. More efforts are required to systematically evaluate the effects of different emission reduction
strategies on alleviating the detrimental effects of $O_3$.

*Data availability.* The source codes of WRF and CMAQ are available at the site
https://www2.mmm.ucar.edu/wrf/users/download/get_sources.html and https://www.cmascenter.org/cmaq/, respectively.
FNL meteorological input files were downloaded from the site https://rda.ucar.edu/datasets/ds083.2/. MEIC v1.3
anthropogenic emission inventory is available at http://meicmodel.org/?page_id=560. The source codes of MEGAN can be
found at https://bai.ess.uci.edu/megan/data-and-code. IAGOS dataset used in model validation was searched and downloaded
from http://iagos-data.fr, which includes all profiles measured in flights taking off from and landing in Hong Kong during
the two representative months. We also provided the initial Fortran code used in ozone budget calculations and hourly $O_3$
concentration and mass budget results in the two representative months (the initial data of Fig. 4) at
https://doi.org/10.5281/zenodo.6259253.

*Author contributions.* KQ, XW and YZ designed the study. KQ, XW, TX did the simulations using the WRF-CMAQ model.
JS, LZ and YZ provided observational results for model validation. KQ, XW, XC, YY, XJ and YZ developed the post-
processing tool *flux_4d_cal*, conducted and analysed $O_3$ budget results. KQ, XW, MV, MK, GB and YZ wrote and/or revised
this paper, with critical feedbacks from all other authors.

*Competing interests.* One of the authors is a member of the editorial board of Atmospheric Chemistry and Physics, and the
peer-review process was guided by an independent editor. The authors declare no other conflict of interest.

*Acknowledgements.* This study was supported by the National Key Research and Development Program of China (grant No.
2018YFC0213204), the National Science and Technology Pillar Program of China (grant No. 2014BAC21B01) and the co-
funded DFG-NSFC Sino-German AirChanges project (grant No. 448720203).

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
