# Peer review of "ozone pollution: Insights from ozone concentration and mass budgets"

_EGUsphere, 2022_

## Author Comment (AC1)

**Response to Reviewer I**

**General Comments:**

- Understanding the processes controlling the $O_3$ concentration in a specific area is important to design emission reduction strategies to reduce the harmful effects of tropospheric $O_3$. This paper focuses on two processes that take place in the $O_3$ cycle: transport and photochemistry.
- The paper discusses two methodological approaches to understand two $O_3$ processes (transport and photochemistry) which are the $O_3$ budget and the $O_3$ source apportionment. Authors claims that there is a contradictory view on the role of transport and photochemistry in $O_3$ pollution between the budget calculation studies and $O_3$ source apportionment studies, because both studies provide different information. In my point of view, they are two different approaches difficult to compare, so it is normal they provide different results. However, as the authors show in the paper it is possible to learn from the both of them.
- I think the paper is organized in a way that it does not help to understand its objective and methodology, even it shows a hard work behind. So, in my opinion, this manuscript was hard to follow and understand, and consequently to review. Furthermore, it could help if authors improve the readability of the text. Overall, there are too many pronouns and missing nouns that make difficult to follow the main idea of some sentences. Authors should review the text carefully and provide a more accurate reference to key concepts, also being consistent in the way they do it along the manuscript.

**Response:**

We appreciate the valuable comments and suggestions. We've tried to adjust the structure of the paper and make a lot of revisions to improve its readability.

Before in-detail responses, we want to clarify the "contradictory" in this paper. Reported $O_3$ concentration budgets often show that photochemistry is the main process leading to the rapid increase of $O_3$ concentrations, but fail to explain why most $O_3$ in the region is transported from the outside regions, as suggested by $O_3$ source apportionment. It indicates that the $O_3$ concentration budget cannot completely illustrate the effects of transport and photochemistry on regional $O_3$ pollution. By calculating, analysing and comparing the $O_3$ concentration and mass budget, this study aims to comprehensively understand the role of transport and photochemistry in regional $O_3$ pollution.

The contents of this paper includes:

1) Development of the method to quantify the two $O_3$ budgets (Sect. 2.1-2.3);

2) Analysis and comparison of the results from the two $O_3$ budgets (methodology described in Sect. 2.5, results discussed in Sect. 3);

3) Assessment of the role of transport and photochemistry in determining the regional origins of $O_3$ (methodology described in Sect. 2.6, result presented in Sect. 4).

Results show that photochemistry dominates the changes of $O_3$ concentrations, or plays a major role in the $O_3$ concentration budget. Although transport only leads to limited changes of $O_3$ concentrations, its large contributions in the $O_3$ mass budget ensure that it determines the characteristics of $O_3$ pollution, e.g., the regional origins of $O_3$ in this study. Based on the conclusions, we suggest the insights from both concentration and mass budgets are necessary to comprehensively understand the role of transport and

chemistry in regional $O_3$ pollution. Suggestions based on the two $O_3$ budgets are also provided for policy-makers when making strategies to alleviate $O_3$ pollution.

Our responses to specific comments and corresponding revisions are as follows (in blue and red, respectively). Note that line numbers are these in the revised manuscript with author's changes.

**Specific comments:**

1) Abstract: difficult to get the important of the problem from the four first lines.

**Response**:

We revised the first four lines in the Abstract as (in lines 20-24):

Understanding the role of transport and photochemistry is essential to mitigate tropospheric ozone ($O_3$) pollution within a region. In previous studies, the $O_3$ concentration budget has been widely used to determine the contributions of two processes to the variations of $O_3$ concentrations. These studies often conclude that local photochemistry is the main cause of regional $O_3$ pollution; however, they fail to explain why $O_3$ in a targeted region is primarily derived from $O_3$ and/or its precursors transported from the outside regions as reported by many studies of $O_3$ source apportionment.

2) The abstract does not help to understands the objective and the methodology approach. Ozone budget calculation and $O_3$ source apportionment studies seems two different type of approaches difficult to compare, so it is normal they provide different results.

**Response:**

We agree that different methods can give different results, but it is also important to know why they are different. For this study, the $O_3$ concentration budget fails to explain why most $O_3$ is transported from the outside regions, suggesting that this method cannot completely illustrate the effects of transport and photochemistry on regional $O_3$ pollution. By calculating, analysing and comparing the $O_3$ concentration and mass budgets, this study not only more comprehensively reveals the role of transport and photochemistry in regional $O_3$ pollution, but also clarifies the connections between $O_3$-related processes and the characteristics of $O_3$, i.e. the regional origins of $O_3$ in this study.

To make a clearer introduction, we revised the objective and the methodology in the abstract, shown in lines 27-32:

Here, we present a method to calculate the hourly contributions of $O_3$-related processes to the variations of not only the mean $O_3$ concentration, but also the total $O_3$ mass (the corresponding budgets are noted as the $O_3$ concentration and mass budget, respectively) within the atmospheric boundary layer (ABL) of the concerned region. Based on the modelling results of WRF-CMAQ, the two $O_3$ budgets were applied to comprehensively understand the effects of transport and photochemistry on the $O_3$ pollution over the Pearl River Delta (PRD) region in China.

3) Line 29: you mention two budgets, but you have not introduced them in the abstract. Is that related with the two type of studies?

**Response:**

Two budgets are introduced in lines 27-31:

Here, we present a method to calculate the hourly contributions of $O_3$-related processes to the variations of not only the mean $O_3$ concentration, but also the total $O_3$ mass (the corresponding budgets are noted as the $O_3$ concentration and mass budget, respectively) within the atmospheric boundary layer (ABL) of the concerned region.

The $O_3$ mass budget is used to explain the results of $O_3$ source apportionment. According to the discussions in Sect. 4 of this paper, transport and photochemistry determine the regional origins of $O_3$ by influencing their contributions in the $O_3$ mass budget as well as the regional origins of $O_3$ mass attributed to these $O_3$-related processes.

4) Line 74: the subject of that sentence "$O_3$ source" does not make sense. Could your elaborate more the idea in that sentence.

**Response:**

We agree that "$O_3$ source" might be a confusing item for the readers. Here, "$O_3$ source" was used to indicate the regional origins of $O_3$, or how much the concerned regions contribute to $O_3$ pollution. We revised the sentence into (in lines 107-111):

$O_3$ source apportionment is performed to identify the regional and/or sectoral origins of $O_3$, of which the results are also used to support air pollution control (Clappier et al., 2017; Thunis et al., 2019). Here, we only discuss the regional origins of $O_3$, because the contributions of sources outside the region (or emissions within the region, defined as local emissions hereafter) provide information on the influence of transport (or photochemistry) on $O_3$ pollution.

We also revised other "$O_3$ sources" in the manuscript into "the results of $O_3$ source apportionment", "regional origins of $O_3$" or alike items.

5) Line 88: "$O_3$ source studies". Use the same set of words to mention these studies. I guess in this case you want to say "$O_3$ source apportionment studies". The same comment in lines 90-91, "source apportionment studies" and "$O_3$ budget studies".

**Response:**

We accept your suggestion. However, the two sentences mentioned in the comment were deleted in the revised version. For the similar expressions afterwards, we revised them into "$O_3$ source apportionment studies", "$O_3$ concentration budget studies" or alike items.

6) Line 93: "CTM are capable of reproducing $O_3$ processes". In this sentence, you are attributing too much credibility to CTM, but models are not perfect and not always reproduce all the processes. I would be more realistic with what CTM can do, so I would suggest to rewrite this sentence.

**Response:**

Thanks for your suggestion. The expression here is inaccurate. In the revised manuscript, this sentence was deleted.

7) Lines 93-103 is specifically to CMAQ, it does not apply to any Eulerian CTMS (i.e. not CTM has a PA module).

**Response:**

Accepted. In the revised manuscript, we pointed out that the method is applied to budget calculations based on WRF-CMAQ results, as shown in lines 173-174:

WRF-CMAQ employs the Process Analysis (PA) module to assess the contributions of $O_3$-related processes to the variations of $O_3$ concentrations within each grid cell.

8) Line 121: "Horizontal transport through the borders of the PRD in four directions". Is that correct? I guess you have two horizontal directions (x and y).

**Response:**

We did not state it clearly. The borders of the PRD were classified as the north, south, west, east border. Horizontal transport through four types of the PRD borders were separately quantified in $O_3$ budgets. Thus, we added more explanations about the classification of border (grid) in lines 199-201:

The PRD grids with one or several interfaces with the outer regions are defined as the border grids, and they can be further classified as the grids in the north, south, west and east borders based on their locations.

and also revised the expressions about the horizontal transport processes in lines 204-205:

The transport processes include horizontal transport through the four types of borders and vertical exchange through the ABL top.

9) Section 2.5 Model setup and validation. Even the model setup is described in the Qu et al. (2021) some basic details should be provided in the text, for example CMAQ and WRF version. Furthermore, the section is named "validation". You mainly referenced Qu et al. (2021) but readers would appreciate a paragraph describing "why" we can trust on your modeling system's results. The evaluation of ABL height with IAGOS measurements is very interesting. Could you elaborate more on the problems with CMAQ during the night?

**Response:**

We agreed that it is necessary to provide more details on model setup, thus relative contents were added in lines 334-352:

The WRF (version 3.2) and CMAQ (version 5.0.2) models were used to simulate the meteorological and pollutant fields, respectively. Two domains with the resolution of 36 and 12 km (denoted as d01 and d02 hereafter) were set up for the one-way nested simulations, and results in the finer d02 were used in the calculations of $O_3$ budgets. To represent the contributions of global background to $O_3$, the initial and

boundary conditions for the coarse d01 domain were provided from the global model, the Model for Ozone and Related Chemical Tracers, version 4 (MOZART-4). The PRD inventory provided by the Guangdong Environmental Monitoring Centre, the Multi-resolution Emission Inventory for China (MEIC) inventory for the mainland China (He, 2012), the MIX inventory for the Asian regions outside of mainland China (Li et al., 2017) and biogenic emissions simulated by the Model of Emissions of Gases and Aerosols from Nature (MEGAN; version 2.10) model were used in the simulations. SAPRC07 (Carter, 2010) and AERO6 were applied as the gas-phase chemistry mechanism and the aerosol scheme, respectively. The simulations of $O_3$ pollution in the PRD were performed for October 2015 (October 11– November 10, 2015) and July 2016 (July 1–31, 2016), which were selected as the representative months in autumn and summer, respectively. Here, $O_3$ polluted days are defined when the maximum hourly $O_3$ concentrations of the day exceed 200 $\mu g/m^3$, or the maximum 8-hour average $O_3$ concentrations of the day exceed 160 $\mu g/m^3$ (both are the Grade-II $O_3$ thresholds in the Chinese National Ambient Air Quality Standard) in any municipality of the PRD. According to this definition, there were 16 and 12 $O_3$ polluted days in the two months, respectively (more information is given in Table S1). The mean $O_3$ budgets during these days were calculated and discussed in the present study.

As for the validation, we agreed that the relative discussions were limited in this part. Thus, we gave more information on:

1) the validation of meteorological parameters, $O_3$, $NO_2$ concentrations and the mixing ratios of hydrocarbons by Qu et al. (2021);

2) the validation of atmospheric boundary layer height, wind speed, direction and ozone mixing ratio at different heights described in detail in Text S3

in a new paragraph, as shown in lines 354-369:

We evaluated the performance of WRF-CMAQ modelling based on multiple observational datasets. The modelling results of meteorological parameters (including temperature, relative humidity and wind speed), $O_3$, $NO_2$ concentrations and the mixing ratios of hydrocarbons were validated with corresponding observations in the PRD by Qu et al. (2021a). The performance of the model to simulate the above variables was overall satisfying with low biases and high correlations (for details, see Qu et al., 2021a). In this study, we further compared the modelled ABL height, the vertical profiles of wind speed, direction and $O_3$ mixing ratio in Hong Kong (located in the south PRD) with the corresponding observations from the IAGOS (In-service Aircraft for a Global Observing System; Petzold et al., 2015) dataset. The modelled ABL heights showed similar hourly variations during the day as the observational results (R = 0.76), with mean bias of -1.1 m (Fig. S2). The mean biases of mean wind speeds are within the range of ± 1 m/s in all height ranges (0-1 km, 1-2 km, 2-5 km), and the results of IAGOS and WRF model indicate similar variations of prevailing wind directions in different seasons and height ranges (Fig. S3). Moreover, modelled $O_3$ mixing ratios in Oct. 2015 are overestimated by 6% and 26% in the height range of 0-1 km and 1-2 km, respectively, and sufficiently illustrate the development, maintenance and dissipation of $O_3$ pollution during the month (Fig. S4). More detailed evaluations on the model performance of these parameters are presented in Text S3 of the Supplement. Overall, the model performance is acceptable, indicating that the model can provide reasonable data for the calculations of $O_3$ budgets.

In this study, we evaluated the modelling performance of atmospheric boundary layer (ABL) height based on the IAGOS potential temperature profiles during daytime, but not at night. One reason is that in Oct.

2015, night-time records are less (30/105 = 28.6%) due to reduced flights at night. Besides, by using potential temperature profiles to determine night-time stable ABL height, large errors may occur (Dai et al., 2014). In order to have more precise $O_3$ budgets, more concerns on night-time ABL height are surely needed in further observations and model validation.

10) Line 220: "acceptable" from which point of view?

**Response:**

In this part, we evaluated the modelling performance of atmospheric boundary layer height, wind speeds, directions and $O_3$ mixing ratios at different heights. The results are summarized as follows:

- The modelled ABL heights showed similar hourly variations during the day as the observational results (R = 0.76), with mean bias of -1.1 m.
- The mean biases of mean wind speeds are within the range of ± 1 m/s in all height ranges (0-1 km, 1-2 km, 2-5 km), and the results of IAGOS and WRF model indicate similar variations of prevailing wind directions in different seasons and height ranges.
- Modelled $O_3$ mixing ratios in Oct. 2015 are overestimated by 6% and 26% in the height range of 0-1 km and 1-2 km, respectively, and sufficiently illustrate the development, maintenance and dissipation of $O_3$ pollution during the month.

High correlations and low biases of these parameters ensures that the modelling results can be used for further analyses, thus they are "acceptable".

According the comment No. 9, relative results are described in the revised manuscript, in lines 354-369.

11) Line 221: "reasonable" from which point of view?

**Response:**

This question is similar as the last one. The good performance of key parameters indicates that the modelling results are close to these in reality, thus they are "reasonable" for further usage in $O_3$ budget calculations.

12) Line 236-237: Is that sentence well written?

**Response:**

We revised the sentence into (in lines 387-391):

The question to be addressed is how $O_3$-related processes determine the regional origins of $O_3$. By combining the $O_3$ mass budget calculations with the BFM source apportionment method, we identified the regional origins of $O_3$ mass changes due to transport and photochemistry (gas-phase chemistry).

13) Source apportion method: Could you comment on the brute force disadvantages for $O_3$ source apportionment calculation? Could CMAQ-ISAM source apportionment method improve your results?

**Response:**

For this part of the study, the goal is to identify the regional origins in the $O_3$ mass changes attributed to transport and gas-phase chemistry (photochemistry). Besides the base scenario, three sensitivity scenarios need to be simulated in the Brute Force Method (BFM), which means increased simulation cost. But the regional source contributions in the $O_3$ mass changes attributed by non-transport processes, including gas-phase chemistry (photochemistry), can be identified. As a tagging method, the ISAM module in CMAQ can be used to identify the regional origins in the $O_3$ mass changes attributed to transport by using $O_3$ concentrations contributed by various regions in calculations. The simulation costs can be reduced, since it is not needed to simulate three sensitivity scenarios. However, as far as we acknowledge, the results for gas-phase chemistry (photochemistry) cannot be provided by the ISAM.

14) Conclusions: "This study concluded that transport and gas-phase chemistry play the main role in the $O_3$ concentration and mass budgets". Is it not new, right? Could you elaborate more this sentence as the main conclusion of this work.

**Response:**

Main conclusions of this study are given in the first paragraph of Sect. 5. This paragraph aims to discuss the application of $O_3$ budgets in the practice of $O_3$ pollution control. As the first sentence, this sentence fails to start the afterward discussions, thus was revised as (in lines 726-728):

The present study concluded that transport and gas-phase chemistry play the main role in the $O_3$ mass and concentration budgets, respectively. As a consequence of our assessment, what should policy-makers do to effectively alleviate regional $O_3$ pollution?

15) Conclusions: Could you elaborate more in the biases in your modelling results? For example, discussing the uncertainties in your emission data, meteorological fields, chemical and meteorological boundary conditions, chemistry in the models.

**Response:**

Emissions, meteorological fields, chemical and meteorological boundary conditions, chemistry and many other factors in models could all influence the results of two $O_3$ budgets. However, this study focuses on the comparison between two $O_3$ budgets to provide a complete view on the role of transport and photochemistry in regional $O_3$ pollution. To have more precise $O_3$ budgets, we suggest to conduct more supporting observations and have more comparisons between observational and modelling results. Specifically, the observational and modelling contributions by various $O_3$-related processes in the $O_3$ budgets can be directly compared. Such results are important for further model development because it indicates which process contribute to high uncertainties in $O_3$ modelling. Relative contents are discussed in the Sect. 5, in lines 717-724.

Uncertainty remains in the calculated $O_3$ budgets, which is partly related to the biases in the modelling results. Therefore, supporting observations are essential for future research. Recent progress in observational techniques (Zhao et al., 2021; Zhou et al., 2021) has enabled three-dimensional measurements of meteorological parameters and $O_3$ concentrations with high spatiotemporal resolution and coverage. These data can be used not only for the model validation of key parameters in budget calculations, but also for the comparisons between observation- and modelling-based contributions by

various O₃-related processes in O₃ budgets (Kaser et al., 2017). The comparison of contributions by O₃-related processes is indicative of the main uncertainties in O₃ pollution modelling, and is therefore also important for further model developments.

**Technical corrections:**

1) Line 93: CTM not defined

**Response:**

This sentence containing "CTM" was deleted in the revised manuscript.

2) Line 95: PA module not defined.

**Response:**

Revised accordingly in line 173:

WRF-CMAQ employs the Process Analysis (PA) module to assess the contributions of O₃-related processes…

3) I would suggest used "tropospheric ozone" instead of "ambient O₃" when possible.

**Response:**

We agreed that to avoid confusion with ozone in stratosphere, "tropospheric ozone" is a better term to be used. It was revised accordingly in line 47-49:

Since first recognized as a key contributor to the Los Angeles smog, tropospheric ozone (O₃) pollution has received considerable attentions in many highly populated areas in the world...

Afterwards, "O₃" is directly used for relative discussions.

**Additional statement:**

Due to their strong professionalism in the areas of atmospheric pollution and modelling as well as high involvement in revising this paper, we are honoured to add Maria Kanakidou and Guy Brasseur as co-authors of this paper.

**References**

Carter, W. P. L.: Development of the SAPRC-07 chemical mechanism, Atmos. Environ., 44, 5324–5335, https://doi.org/10.1016/j.atmosenv.2010.01.026, 2010.

Clappier, A., Belis, C. A., Pernigotti, D., and Thunis, P.: Source apportionment and sensitivity analysis: two methodologies with two different purposes, Geosci. Model Dev., 10, 4245–4256, https://doi.org/10.5194/gmd-10-4245-2017, 2017.

Dai, C., Wang, Q., Kalogiros, J. A., Lenschow, D. H., Gao, Z., and Zhou, M.: Determining Boundary-Layer Height from Aircraft Measurements, Bound.-Lay. Meteorol., 152, 277–302, https://10.1007/s10546-014-9929-z, 2014.

He, K.: Multi-resolution Emission Inventory for China (MEIC): model framework and 1990-2010 anthropogenic emissions, American Geophysical Union, Fall Meeting 2012, 3–7 December 2012, San Francisco, USA, A32B-05, 2012.

Kaser, L., Patton, E. G., Pfister, G. G., Weinheimer, A. J., Montzka, D. D., Flocke, F., Thompson, A. M., Stauffer, R. M., and Halliday, H. S.: The effect of entrainment through atmospheric boundary layer growth on observed and modeled surface ozone in the Colorado Front Range, J. Geophys. Res.-Atmos., 122, 6075–6093, https://doi.org/10.1002/2016JD026245, 2017.

Li, M., Zhang, Q., Kurokawa, J.-I., Woo, J.-H., He, K., Lu, Z., Ohara, T., Song, Y., Streets, D. G., Carmichael, G. R., Cheng, Y., Hong, C., Huo, H., Jiang, X., Kang, S., Liu, F., Su, H., and Zheng, B.: MIX: a mosaic Asian anthropogenic emission inventory under the international collaboration framework of the MICS-Asia and HTAP, Atmos. Chem. Phys., 17, 935–963, https://doi.org/10.5194/acp-17-935-2017, 2017.

Petzold, A., Thouret, V., Gerbig, C., Zahn, A., Brenninkmeijer, C. A. M., Gallagher, M., Hermann, M., Pontaud, M., Ziereis, H., Boulanger, D., Marshall, J., Nédélec, P., Smit, H. G. J., Friess, U., Flaud, J.-M., Wahner, A., Cammas, J.-P., Volz-Thomas, A. and IAGOS TEAM: Global-scale atmosphere monitoring by in-service aircraft–current achievements and future prospects of the European Research Infrastructure IAGOS, Tellus B, 67, 28452, https://doi.org/10.3402/tellusb.v67.28452, 2015.

Qu, K., Wang, X., Yan, Y., Shen, J., Xiao, T., Dong, H., Zeng, L., and Zhang, Y.: A comparative study to reveal the influence of typhoons on the transport, production and accumulation of $O_3$ in the Pearl River Delta, China, Atmos. Chem. Phys., 21, 11593–11612, https://doi.org/10.5194/acp-21-11593-2021, 2021.

Thunis, P., Clappier, A., Tarrason, L., Cuvelier, C., Monteiro, A., Pisoni, E., Wesseling, J., Belis, C. A., Pirovano, G., Janssen, S., Guerreiro, C., and Peduzzi, E.: Source apportionment to support air quality planning: Strengths and weaknesses of existing approaches, Environ. Int., 130, 104825, https://doi.org/10.1016/j.envint.2019.05.019, 2019.

Zhao, R., Hu, Q., Sun, Z., Wu, Y., Xing, C., Liu, H., and Liu, C.: Review of space and ground integrated remote sensing for air pollutants (in Chinese). Res. Environ. Sci., 34(1), 28-40. https://doi.org/10.13198/j.issn.1001-6929.2020.11.25, 2021.

Zhou, B., Zhang, S., Xue, R., Li, J., and Wang, S.: A review of Space-Air-Ground integrated remote sensing techniques for atmospheric monitoring, J. Environ. Sci., https://doi.org/10.1016/j.jes.2021.12.008, 2021.

---

## Author Comment (AC2)

**Response to Reviewer II**

**General Comments:**

Qu et al. present an analysis of the $O_3$ budget in the ABL in two different ways: a concentration budget and a mass budget. They apply the budget calculations to the $O_3$ budget over the Pearl River Delta based on simulations with WRF-CMAQ. The 2 different ways of calculating the $O_3$ budget lead to opposing views on the main contributions to the $O_3$ budget: while photochemistry dominates in the concentration budget, (vertical) transport dominates the mass budget. A tool is developed to calculate the budget contributions. A control simulation is performed, and in addition 3 brute force emission reduction scenarios are carried out. Budget calculation following the 2 methods are performed and the differences discussed.

Unfortunately, the way the paper is written makes it hard to judge its scientific merits, and I cannot recommend acceptation in its current form.

**Response:**

We appreciate the valuable comments and suggestions. We've tried to adjust the structure of the paper and make a lot of revisions to improve its readability.

Our responses to specific comments and corresponding revisions are as follows (in blue and red, respectively). Note that line numbers are these in the revised manuscript with author's changes.

**Major comments:**

1) This is a dense paper without much guidance for the reader as to where you are going, which makes it hard to follow, and hard to judge the scientific merits of the work you describe. I had to reread it 3 times and still I am getting lost in the details. Please rewrite it in a more structured way, and indicate the purpose of each section in its first sentence. For instance, in section 2.6 a number of scenario runs seems to appear out of the blue. Where are the results of these runs used/discussed?

**Response:**

Thanks for the suggestions. We have revised the manuscript and made it more structured, clear and reader-friendly. Pointing out the purpose of each section is surely a good way to provide readers more clues in reading — we have applied this suggestion in the revisions.

The basic logic of this paper is as follows. The objective is to comprehensively illustrate the effects of transport and photochemistry on regional $O_3$ pollution from the perspectives of both $O_3$ concentration and mass budgets. Three tasks are included in this study:

1) Development of the method to quantify the two $O_3$ budgets (Sect. 2.1-2.3);

2) Analysis and comparison of the results from the two $O_3$ budgets (methodology described in Sect. 2.5, results discussed in Sect. 3);

3) Assessment of the role of transport and photochemistry in determining the regional origins of $O_3$ (methodology described in Sect. 2.6, result presented in Sect. 4).

In the introduction part, we re-wrote the relevant paragraphs to overview the structure of this manuscript, as shown in lines 157-192:

[revised manuscript text omitted]

which separately discuss the results of aforementioned task 2 and 3.

2) What is actually lacking is an explanation of why 2 different budget methods give such different results. Is it mainly a boundary conditions problem? A change in mass does not lead to a change in concentration when the background concentration is similar over larger regions? Maybe it is discussed in L445-448?

**Response:**

The two $O_3$ budgets describe the conservations of $O_3$ concentration and mass in the atmospheric boundary layer (ABL) of the region. As introduced in Sect. 2.4, the different results of two $O_3$ budgets are mainly attributed to the different effects of transport on $O_3$ concentration and mass. When $O_3$ is transported into (or out of) the ABL of the region through the advection process (horizontal transport and vertical exchange through the ABL top due to large-scale air motion (ABLex-M)), surely total $O_3$ mass increases (or decreases). However, whether $O_3$ concentration increases or decreases also depends on the difference between $O_3$ concentrations in the region and transported air parcels — that is why clean (polluted) air parcels being transported into the region dilutes (aggravates) $O_3$ pollution. The effect of transport can be understood as to replace a part of air mass with the transported air parcel. If $O_3$ concentration is higher (or lower) in the transported air parcel, by replacing, mean $O_3$ concentration within the region will increase (or decrease). This effect also applies to the exchange through the ABL top due to the temporal changes of ABL heights (ABLex-H). For example, after sunrise, $O_3$ mass in the ABL of the region increases rapidly along with the development of ABL. This process can be viewed as two air parcels combining into one, and whether $O_3$ concentration increases or decreases also depends on the difference of $O_3$ concentrations in two air parcels — but $O_3$ mass surely increases. More detailed contents are discussed in Sect. 2.4, in lines 301-321:

The difference between the two $O_3$ budgets is linked to the varied effects of transport on $O_3$ mass and concentration. Suppose that the mean $O_3$ concentration in the transported air parcels is $\langle c_{O_3} \rangle_{trans}$. For horizontal transport, its contributions in the $O_3$ mass and concentration budgets can be separately written as:

$$F_{htrans} = \langle c_{O_3} \rangle_{trans}\, dV \tag{8}$$

$$d\langle c_{O_3} \rangle_{htrans} = \frac{dV}{V}\left( \langle c_{O_3} \rangle_{trans} - \langle c_{O_3} \rangle \right) \tag{9}$$

Apparently, $F_{htrans}$ is related to the $O_3$ concentrations in the transported air parcels, but not to those in the studied region. It indicates how much $O_3$ is transported into or out of the region. Whether it is positive or negative only depends on the direction of transport — $O_3$ being transported into (out of) the region leads to the increase (decrease) of $O_3$ mass, which corresponds to a positive (negative) contribution in the $O_3$ mass budget. In contrast, $d\langle c_{O_3}\rangle_{htrans}$ quantifies how much horizontal transport alters regional-mean $O_3$ concentrations, and is linked to the difference between $O_3$ concentrations in the transported air parcels and the studied region (Eq. (9)). $O_3$ being transported into (out of) the region does not necessarily result in a higher (lower) $O_3$ concentration. For instance, when clean air parcels with relatively low $O_3$ levels are transported into the region, they dilute $O_3$ pollution and reduce $O_3$ concentration ($d\langle c_{O_3}\rangle_{htrans} < 0$). Given that ABLex-M is also an advection process, the above difference applies to this process as well. For ABLex-H, its contributions in the $O_3$ mass and concentration budgets are expressed as:

$$F_{ABLex-H} = \langle c_{O_3}\rangle_{trans}\, dV \tag{10}$$

$$d\langle c_{O_3}\rangle_{ABLex-H} = \frac{dV}{V + dV}\left(\langle c_{O_3}\rangle_{trans} - \langle c_{O_3}\rangle\right) \tag{11}$$

Similarly, ABL development and collapse lead to the increase and decrease of $O_3$ mass, respectively, but whether they contribute to higher or lower $O_3$ concentration also depends on the difference between $O_3$ concentration in the transported air parcels and that in the region. Based on the above discussion, these transport processes all show different effects on $O_3$ mass and concentration — the effect of transport on the variations of $O_3$ mass is only related to the characteristics of the transported air parcels, namely their volumes and $O_3$ concentrations within (Eqs. (8) and (10)), while how transport contributes to the variations of $O_3$ concentration is linked to the difference between $O_3$ concentrations in the transported air parcels and the region (Eqs. (9) and (11)).

It is possible that change in mass does not lead to a change in concentration when the background concentration is similar over larger regions. For example, suppose that air parcels with the volume of $dV$ and the $O_3$ concentration of $\langle c_{O_3}\rangle_{trans}$ are transported into the region through the ABLex-H process. The contributions of such a process to $O_3$ mass and concentration (denoted as $F_{ABLex-H}$ and $d\langle c_{O_3}\rangle_{ABLex-H}$, respectively) can be expressed as (Eqs. (10-11) of the manuscript):

$$F_{ABLex-H} = \langle c_{O_3}\rangle_{trans}\, dV$$

$$d\langle c_{O_3}\rangle_{ABLex-H} = \frac{dV}{V + dV}\left(\langle c_{O_3}\rangle_{trans} - \langle c_{O_3}\rangle\right)$$

where $V$ is the original volume of the ABL of the region, $\langle c_{O_3}\rangle$ is the initial mean $O_3$ concentration. $F_{ABLex-H}$ is surely positive, since $\langle c_{O_3}\rangle_{trans}$ and $dV$ are both above 0. As an extreme case, if $\langle c_{O_3}\rangle_{trans} = \langle c_{O_3}\rangle$, then $d\langle c_{O_3}\rangle_{ABLex-H} = 0$, which means that the transport, or "combination", of air parcels with the same $O_3$ concentration leads to increased $O_3$ mass and volume in the ABL of the region at the same time, but $O_3$ concentration does not change.

In L445-448 of the original manuscript, we discussed why the conclusions in this paper are important for further studies. Differences in the concentration and mass budgets apply to not only $O_3$, but also other pollutants with moderately long atmospheric lifetimes, such as fine particulate matter and some of its components. Transport may fail to notably alter pollutant concentration, but can significantly contribute to the changes of pollutant mass. Specifically, massive pollutant being transported into the ABL in the morning nearly determines the characteristics of pollutant within the region — besides the origins of

pollutants, they also include the contributions of different reaction pathways and sensitivities to precursor emissions. But in the concentration budget, the effects of transport on these characteristics are often ignored. In order to fully understand the effects of transport, chemistry and other related processes, we suggest that the insights from both concentration and mass budgets are required for future studies.

**Minor comments:**

1) L50 (and throughout MS): $O_3$ processes --> $O_3$-related processes

**Response:**

Accepted and revised as suggested.

2) L74: pls rephrase sentence

**Response:**

This sentence is revised into (in lines 107-109):

$O_3$ source apportionment is performed to identify the regional and/or sectoral origins of $O_3$, of which the results are also used to support air pollution control (Clappier et al., 2017; Thunis et al., 2019).

3) L416: "High contributions of …" Unclear sentence. Please rephrase.

**Response:**

This sentence is revised into (in lines 650-654):

By combining the $O_3$ mass budget and $O_3$ source apportionment, we identified the $O_3$ mass increase due to $O_3$-related processes as local (PRD) and non-local (EC-China and BCON) contributions. According to the results discussed before, high contributions of transport in the morning-hour $O_3$ mass increase and the dominance of non-local source contributions in this part of new $O_3$ ensure that non-local sources contributed to most $O_3$ in the PRD.

4) L461: what do you mean by 'a longer time'?

**Response:**

 "A longer time" is vague, thus it is revised as in lines 732-733:

However, for short-term air pollution control, this strategy is not efficient because emission reduction in upwind regions may need to start days earlier before the polluted periods.

**Additional statement:**

Due to their strong professionalism in the areas of atmospheric pollution and modelling as well as high involvement in revising this paper, we are honoured to add Maria Kanakidou and Guy Brasseur as co-authors of this paper.

---

## Referee Report (RR1)

Qu et al. have presented and nuanced two different types of budgets for regional ozone pollution: concentration budget and mass budget. They start by mathematically formulating both types of budgets from first principles, basically using the fundamental principle of mass conservation in a Eulerian framework, similar to a continuity equation where the rate of change of concentration/mass is the sum of horizontal advection, vertical exchange and local source term (chemical production, dry deposition etc.). They then present their method for calculating the two types of budgets on WRF-CMAQ gridded model output. They have chosen the Pearl River Delta region (PRD) in China as their study region.

For the concentration budget, they break down the vertical exchange term into two separate terms: vertical entrainment/detrainment of air due to temporal changes in boundary layer height (ABLex-H) and horizontal advection of air through the extra volume of air created due to increasing boundary layer height (ABLex-M). Since the study region is not a perfect cube, they have defined four boundaries roughly corresponding to north, east, west and south directions, like four sides of a cube, to deal with the transport in a Eulerian framework. To calculate the transport contribution to the change in concentration in the boundary layer, they make use of the concentrations of the horizontal advecting air mass as well as the background concentration of the air above the boundary layer in the region. Similarly, transport contribution in the mass budget is calculated by adding the new mass brought in through advection and vertical exchange.

The key point here is that often new mass is added from non-local sources through transport but this increase in mass is simultaneously accompanied by an increase in boundary layer volume which diminishes any considerable increase in concentration within the boundary layer. Therefore, these non-local contributions are diminished in the concentration budget although the composition of the pollution has changed, i.e., there are more O3 molecules in the region from the non-local sources without any (or much) change in concentration.

The authors then perform 3 different sensitivity simulations where they zero-out emissions for the PRD region, Eastern and Central China region but not PRD, and all regions within inner model domain, respectively. Using the difference between these sensitivity simulations and the baseline run, they calculate the contributions of these source regions to the O3 mass and O3 concentration in the PRD region. In Figure 3 they show that the change in mass is driven in large amount by vertical entrainment but this addition of transported mass in the morning is accompanied by increase in boundary layer volume and the removal of transported mass in the evening is accompanied by a decrease in boundary volume such that the transport does not have a large effect on concentration budget. They further show in Figure 6 that a major part of vertical exchange and horizontal transport in the mass budget comes from non-local and background sources, and that the horizontal transport is greater than local chemical production in autumn and the opposite in summer.

Overall, the authors have highlighted an important point on mass contributions of O3 (or any other longer-lived pollutant) which gets concealed in concentration budgets due to volume changes in boundary layer. Mass budget might become more important than concentration budget particularly in cases when the chemical species in consideration has a different characteristic (e.g., toxicity) based on its source region. I recommend this manuscript for publication with minor corrections:

1. Include the domain map showing at least d02 of WRF-CMAQ with clear demarcation of the different source regions used in the BFM simulations.
2. The names ABLex-H and ABLex-M aren't intuitive. I do not understand why those letters (H and M) were used as they can confuse the reader. I suggest calling them ABLex-A (advection through boundary layer change) and ABLex-E (entrainment through boundary layer change).

3. In this work, the authors have formulated their equations to calculate "change" in concentration and mass over time but there are plenty of studies which perform BFM-type sensitivity runs where they alter emissions over different regions and subtract the result from the baseline run to derive hourly concentrations (instead of hourly change in concentrations) attributed to emissions from that region. The authors should discuss the validity of such results and their implications for policymaking.

---

## Author Response (AR2)

**Response to Reviewer III**

**General Comments:**

Qu et al. have presented and nuanced two different types of budgets for regional ozone pollution: concentration budget and mass budget. They start by mathematically formulating both types of budgets from first principles, basically using the fundamental principle of mass conservation in a Eulerian framework, similar to a continuity equation where the rate of change of concentration/mass is the sum of horizontal advection, vertical exchange and local source term (chemical production, dry deposition etc.). They then present their method for calculating the two types of budgets on WRF-CMAQ gridded model output. They have chosen the Pearl River Delta region (PRD) in China as their study region.

For the concentration budget, they break down the vertical exchange term into two separate terms: vertical entrainment/detrainment of air due to temporal changes in boundary layer height (ABLex-H) and horizontal advection of air through the extra volume of air created due to increasing boundary layer height (ABLex-M). Since the study region is not a perfect cube, they have defined four boundaries roughly corresponding to north, east, west and south directions, like four sides of a cube, to deal with the transport in a Eulerian framework. To calculate the transport contribution to the change in concentration in the boundary layer, they make use of the concentrations of the horizontal advecting air mass as well as the background concentration of the air above the boundary layer in the region. Similarly, transport contribution in the mass budget is calculated by adding the new mass brought in through advection and vertical exchange.

The key point here is that often new mass is added from non-local sources through transport but this increase in mass is simultaneously accompanied by an increase in boundary layer volume which diminishes any considerable increase in concentration within the boundary layer. Therefore, these non-local contributions are diminished in the concentration budget although the composition of the pollution has changed, i.e., there are more $O_3$ molecules in the region from the non-local sources without any (or much) change in concentration.

The authors then perform 3 different sensitivity simulations where they zero-out emissions for the PRD region, Eastern and Central China region but not PRD, and all regions within inner model domain, respectively. Using the difference between these sensitivity simulations and the baseline run, they calculate the contributions of these source regions to the $O_3$ mass and $O_3$ concentration in the PRD region. In Figure 3 they show that the change in mass is driven in large amount by vertical entrainment but this addition of transported mass in the morning is accompanied by increase in boundary layer volume and the removal of transported mass in the evening is accompanied by a decrease in boundary volume such that the transport does not have a large effect on concentration budget. They further show in Figure 6 that a major part of vertical exchange and horizontal transport in the mass budget comes from non-local and background sources, and that the horizontal transport is greater than local chemical production in autumn and the opposite in summer.

Overall, the authors have highlighted an important point on mass contributions of $O_3$ (or any other longer-lived pollutant) which gets concealed in concentration budgets due to volume changes in boundary layer. Mass budget might become more important than concentration budget particularly in cases when the chemical species in consideration has a different characteristic (e.g., toxicity) based on its source region. I recommend this manuscript for publication with minor corrections.

**Response:**

We appreciate your positive comments on our paper. Following your summary of our contents in the general comments, for clarity, we have modified the abstract, in lines 33-38 (the line numbers used correspond to those in the revised manuscript with author's changes; same below):

Through high contributions to the $O_3$ mass increase in the morning, transport determines that most $O_3$ in the PRD originates from the global background and emissions outside the region. However, due to the simultaneous rapid increase of ABL volumes, this process only has a relatively limited effect on $O_3$ concentration increase compared to photochemistry, and transport effect on the regional sources of $O_3$ cannot be illustrated by the $O_3$ concentration budget.

and the revised manuscript, in lines 551-553:

However, accompanied with the simultaneous rapid increase of ABL volumes, this process has a relatively limited contribution to $O_3$ concentration increase in comparison to photochemistry.

and in lines 574-578:

Massive $O_3$, mostly derived from non-local sources, being transported into the ABL in the morning has a relatively limited influence on the $O_3$ concentration increase (25% and 5% in autumn and summer, respectively) compared to photochemistry because of the rapid change of ABL volumes at the same time. However, this process nearly determines the dominance of non-local source contributions for daytime $O_3$ in the PRD.

The manuscript has also been revised based on other suggestions. Please find below our responses to the specific comments (in blue) and corresponding revisions (in red).

**Specific comments:**

1) Include the domain map showing at least d02 of WRF-CMAQ with clear demarcation of the different source regions used in the BFM simulations.

**Response**:

In the original manuscript, the domain map was displayed as Fig. S5 in Supplement. We agreed that such information might be important for readers. Thus, it is now shown as the new Fig. 2 in the manuscript. All figure numbers have been corrected accordingly.

2) The names ABLex-H and ABLex-M aren't intuitive. I do not understand why those letters (H and M) were used as they can confuse the reader. I suggest calling them ABLex-A (advection through boundary layer change) and ABLex-E (entrainment through boundary layer change).

**Response:**

We accept the suggestion and try to express these processes in a more reader-friendly way:

**The vertical exchange near the ABL top due to large-scale air motion** is a process due to the advection perpendicular to the ABL top and its slope. We agree that "ABLex-A" is a better short term to indicate the process, thus all "ABLex-M" in the manuscript, tables and figures were revised into "ABLex-A". Besides, in the full term, "due to large-scale air motion" may not be clear for readers, thus it has been revised into "due to advection perpendicular to the ABL top and its slope" in the manuscript.

**The vertical exchange near the ABL top due to the changes in ABL heights** occurs only linked to the increases and decreases of ABL heights. It is a part of the process of vertical exchange near the ABL top, or entrainment and detrainment. Thus, we prefer to keep the short term of the process as "ABLex-H", where "H" in this manuscript is used as the parameter of ABL height in this paper. We added the note to indicate ABL height in the full name of the process is represented by "H" before introducing the short term, in lines 155-156:

… 1) the temporal changes of ABL heights ($H$) and 2) …

It is also needed to clarify that vertical exchange near the ABL top is the process of entrainment and detrainment. Thus, we added some necessary notes in relative parts, including lines 74-75:

…2) vertical exchange through the ABL top (entrainment and detrainment, the third term) …

 and lines 154-155:

The terms on the right side of Eq. (3) suggest that vertical exchange through the ABL top, or entrainment and detrainment, is attributed to …

3) In this work, the authors have formulated their equations to calculate "change" in concentration and mass over time but there are plenty of studies which perform BFM-type sensitivity runs where they alter emissions over different regions and subtract the result from the baseline run to derive hourly concentrations (instead of hourly change in concentrations) attributed to emissions from that region. The authors should discuss the validity of such results and their implications for policymaking.

**Response:**

We thank the reviewer for the comment. To better clarify: It describes the application of BFM, a typical method of $O_3$ source apportionment, which aims to identify the contributions of emissions originating from different regions to the $O_3$ level in the targeted region. Indeed, there are several BFM studies (e.g., Wang et al. (2006) and Streets et al. (2007)) that are used for policymaking.

Our study does not question the validity of such $O_3$ source apportionment studies or $O_3$ concentration budget analysis, but suggests appropriate ways on how to apply these methods effectively: i) To lower the overall $O_3$ levels and achieve long-term air quality improvement, based on the results of $O_3$ source apportionment, it is needed to focus on emission reduction within larger areas for regions that are notably influenced by upwind sources; ii) To lower the peak $O_3$ levels of the day and achieve short-term alleviation of $O_3$ pollution, owing to the quick response of $O_3$ concentration increase to local emissions in the $O_3$ concentration budget, reducing local emissions is a better strategy. The choice of strategy to apply should depend on the specific goal of $O_3$ pollution control, which is set based on the effects of $O_3$ pollution on human health, the ecosystem, etc. Relative discussions can be found of in the final paragraph of this manuscript, in lines 603-614:

The present study concluded that transport and gas-phase chemistry play the main role in the $O_3$ mass and concentration budgets, respectively. As a consequence of our assessment, the following is suggested for policy-makers. For areas where non-local emissions notably contribute to $O_3$, emission reduction in the upwind regions can reduce the overall $O_3$ concentrations effectively, which is a crucial step towards the long-term improvement of regional air quality. However, for short-term air pollution control, this strategy is not efficient because emission reduction in upwind regions may need to start days earlier before the polluted periods. In contrast, reducing local emissions is expected to lower the rapid daytime $O_3$

concentration increase efficiently and, thereby, $O_3$ peak levels in the short term, as highlighted by the $O_3$ concentration budget. The choice of the better strategy to be applied should depend on the specific objectives of $O_3$ control (mean levels vs. peak levels; long-term vs. short-term), which are set based on a more in-depth understanding of $O_3$ effects on human health, crop yields and ecosystems. More efforts are required to systematically evaluate the effects of different emission reduction strategies on alleviating the detrimental effects of $O_3$.

**Additional Statement:**

During the validation of the revised manuscript, the ACP team noted that:

*Please make sure that the lists of corresponding authors in the system and the manuscript file match.*

The corresponding authors of this paper are Xuesong Wang and Yuanhang Zhang. However, in the submitting system, Xiao Teng was automatically assigned as the corresponding author and we cannot cancel it. Please note that this is not correct and need to be corrected.

**Reference**

Streets, D. G., Fu, J. S., Jang, C. J., Hao, J., He, K., Tang, X., Zhang, Y., Wang, Z., Li, Z., Zhang, Q., Wang, L., Wang, B., Yu, C., Air Quality during the 2008 Beijing Olympic Games, Atmos. Environ., 41(3), 480–492, https://doi.org/10.1016/j.atmosenv.2006.08.046, 2007.

Wang, Z., Li, J., Wang, X., Pochanart, P., Akimoto, H., Modeling of Regional High Ozone Episode Observed at Two Mountain Sites (Mt. Tai and Huang) in East China, J. Atmos. Chem., 55(3), 253–272, https://doi.org/10.1007/s10874-006-9038-6, 2006.